# SPIN enables high throughput species identification of archaeological bone by proteomics

Patrick Leopold Rüther [1✉], Immanuel Mirnes Husic[1], Pernille Bangsgaard[2], Kristian Murphy Gregersen[3], Pernille Pantmann[4], Milena Carvalho[5,6], Ricardo Miguel Godinho[5], Lukas Friedl[5,7], João Cascalheira[5], Alberto John Taurozzi [2], Marie Louise Schjellerup Jørkov[8], Michael M. Benedetti [5,9], Jonathan Haws [5,10], Nuno Bicho [5], Frido Welker[2], Enrico Cappellini [2] & Jesper Velgaard Olsen [1✉]

Species determination based on genetic evidence is an indispensable tool in archaeology, forensics, ecology, and food authentication. Most available analytical approaches involve compromises with regard to the number of detectable species, high cost due to low throughput, or a labor-intensive manual process. Here, we introduce "Species by Proteome INvestigation" (SPIN), a shotgun proteomics workflow for analyzing archaeological bone capable of querying over 150 mammalian species by liquid chromatography-tandem mass spectrometry (LC-MS/MS). Rapid peptide chromatography and data-independent acquisition (DIA) with throughput of 200 samples per day reduce expensive MS time, whereas streamlined sample preparation and automated data interpretation save labor costs. We confirm the successful classification of known reference bones, including domestic species and great apes, beyond the taxonomic resolution of the conventional peptide mass fingerprinting (PMF)-based Zooarchaeology by Mass Spectrometry (ZooMS) method. In a blinded study of degraded Iron-Age material from Scandinavia, SPIN produces reproducible results between replicates, which are consistent with morphological analysis. Finally, we demonstrate the high throughput capabilities of the method in a high-degradation context by analyzing more than two hundred Middle and Upper Palaeolithic bones from Southern European sites with late Neanderthal occupation. While this initial study is focused on modern and archaeological mammalian bone, SPIN will be open and expandable to other biological tissues and taxa.

[1] Novo Nordisk Foundation Center for Protein Research, University of Copenhagen, Copenhagen, Denmark. [2] Globe institute, University of Copenhagen, Copenhagen, Denmark. [3] Institute of Conservation, Royal Danish Academy, Copenhagen, Denmark. [4] Dept. of Archaeology, Museum Nordsjælland, Copenhagen, Denmark. [5] Interdisciplinary Center of Archaeology and Evolution of Human Behavior, University of Algarve, Faro, Portugal. [6] Departmrent of Anthropology, University of New Mexico, Albuquerque, NM, USA. [7] Dept. of Anthropology University of West Bohemia, Pilsen, Czech Republic. [8] The Laboratory of Biological Anthropology, Department of Forensic Medicine, University of Copenhagen, Copenhagen, Denmark. [9] Department of Earth and Ocean Sciences, University of North Carolina Wilmington, Wilmington, NC, USA. [10] Department of Anthropology, University of Louisville, Louisville, KY, USA. ✉email: patrick.ruether@palaeome.org; jesper.olsen@cpr.ku.dk

olecular identification methods enable species determination of archaeological objects and other biological samples independent of morphological features. Genetic methods analyze species-specific variations either in DNA, which provides maximum resolution[1,2], or in proteins, which have superior longevity and can be analyzed without amplification bias[3,4]. The two most common approaches to protein-based species identification are immuno-assays for targeting a single or few selected species at a time[5] and mass spectrometry (MS) for global species determination[6,7]. Here, we employ MS-based methods because they offer greater flexibility and can cope with more degraded and contaminated material[8,9].

The typical workflow for MS-based protein analysis involves tryptic digestion of extracted proteins into shorter peptides, which are subsequently analyzed by liquid chromatography-tandem mass spectrometry (LC-MS/MS)[10] or by peptide mass fingerprinting (PMF)[11]. Peptide sequencing by LC-MS/MS using high-resolution MS instrumentation facilitates confident identification of proteins in a complex biological matrix and is therefore the standard technology in proteomics. The technique is broadly compatible with virtually any type of protein and suitable for single and mixed species analysis. The recent development of high-throughput data acquisition and interpretation workflows for clinical diagnostics[12,13] have yet to be adapted with regard to the samples and research questions relevant for molecular species identification. As of now, species identification by LC-MS/MS is primarily reserved for a few individual samples of high profile that justify the high analytical costs, long analysis time, and complex data interpretation[14,15]. Conversely, most studies with many samples continue to be based on the faster and more cost-efficient PMF[16,17]. Here, peptides are analyzed without chromatographic separation by matrix-assisted laser desorption/ionization time-of-flight (MALDI-TOF) MS, which delivers high throughput but has limited versatility and dynamic range, and suffers from lower confidence in peptide identifications due to a lack of MS/MS-based sequencing. The downside of this is that the samples cannot be too complex and should contain, or be dominated by, only one or few proteins, such as type I collagen (Collagen alpha-1(I) chain and Collagen alpha-2(I) chain) in bone. A popular PMF-based workflow for collagen-based species identification is "Zooarchaeology by Mass Spectrometry" (ZooMS)[18,19]. Although applicable to many groups of vertebrates, the maximum obtainable species resolution remains limited by the relatively small number of analyzed peptides from just two evolutionary conserved proteins. Consequently, there are many species that are indistinguishable by ZooMS, in particular within closely related genera. Unsurprisingly, their bones often share a highly similar morphology and therefore frequently require molecular species identification for unambiguous classification.

With "Species by Proteome INvestigation" (SPIN), we present an LC-MS/MS-based species identification workflow that combines sequencing-based reliability from conventional LC-MS/MS with the speed and cost-effectiveness from ZooMS, ultimately overcoming the current limits of both these two approaches. To facilitate the analysis of large cohorts with hundreds to thousands of samples, major bottlenecks in sample preparation, data acquisition, and conclusive species assignment were removed. While the SPIN concept can in principle be applied to many protein-containing materials, we selected bone remains for its first implementation, due to their high relevance in archaeology and forensics.

Preparing hundreds of samples per day requires a scalable, reproducible, and automated procedure with few manual steps resulting in peptides without contaminants that could interfere with MS analysis. To achieve this, we took advantage of the protein aggregation capture (PAC) cleanup and digestion method using magnetic beads[20,21] that have previously been applied to

bone proteins[22,23]. PAC can be transferred to a robotics platform[24] and its robustness allowed us to use a combined bone-demineralization/protein-extraction solution and thereby omit buffer exchange, a time-consuming step during which significant protein loss can occur. Although this protocol is not necessarily faster than standard in-solution digestion[25] for a single sample, the automation and protein-level cleanup facilitate the scalability needed for routine high-throughput analysis.

The cost of high-resolution tandem mass spectrometers poses a major barrier to the application of conventional LC-MS/MS for large-scale species identification. Instead of compromising on technology, we aimed at reducing the cost per sample by shortening the acquisition time by an order of magnitude compared to previous LC-MS/MS-based species identification strategies. This became possible due to two recent developments, liquid chromatography (LC) systems with significantly shorter overhead time between analyses[26] and software for the confident peptide identification based on data-independent acquisition (DIA)[27,28]. In addition to higher throughput and lower analysis costs, shorter gradients are beneficial for sensitivity and the Evosep LC makes tedious preparation steps like peptide elution after cleanup, solvent evaporation, and MALDI target plate spotting obsolete. As an alternative to using the latest LC-MS/MS equipment, the data acquisition methods required for SPIN should theoretically be transferable to previous generations of Orbitrap MS instruments[29,30] and lower sensitivity microflow LC systems[31] if compensated with higher peptide loads.

The interpretation of the peptide identification data to determine the most probable species is an effort that is often neglected when comparing different technologies. Particularly in large projects with optimized and parallelized laboratory processes, downstream data interpretation is often the major bottleneck if done manually. Nevertheless, this has been the case for virtually all species identification studies based on proteins[15,32], even despite the development of a software tool for species analysis by PMF of collagen[33] that has unfortunately not been made publicly available. In the SPIN workflow, peptides are identified using broadly available and confidence-controlled peptide search software, such as Spectronaut[27] and Maxquant[34], followed by data interpretation by a newly developed automated species inference algorithm that rationally compares the MS results with a species database by sequence alignment. In contrast to the previously described tool for ZooMS[33], our algorithm operates without untraceable machine learning steps and does not strictly require experimental spectral libraries.

We optimized our methods with reference bones from nine domesticated animals, human, and three other great ape species, validated the reproducibility, and benchmarked comparability to morphology-based identification with more than 60 partially degraded bones from the Danish Early Iron-Age. Finally, we stress-tested the SPIN approach with a set of over 200 Middle and Upper Palaeolithic bone fragments from three archaeological sites in Portugal dating to 30–60,000 BP. We demonstrate that mammalian species families and taxa can be correctly assigned in most cases, while misassignments due to low signal intensity or gaps in the species sequence database can be avoided through thresholds based on two different quality control confidence scores. Comparative analysis of the same peptides from the references and a subset of the archaeological bones by PMF-based ZooMS analysis confirmed superior separation of closely related taxa and higher sensitivity with SPIN.

## Results

**Streamlined and automated sample preparation.** To increase sample processing throughput, we developed a sample preparation

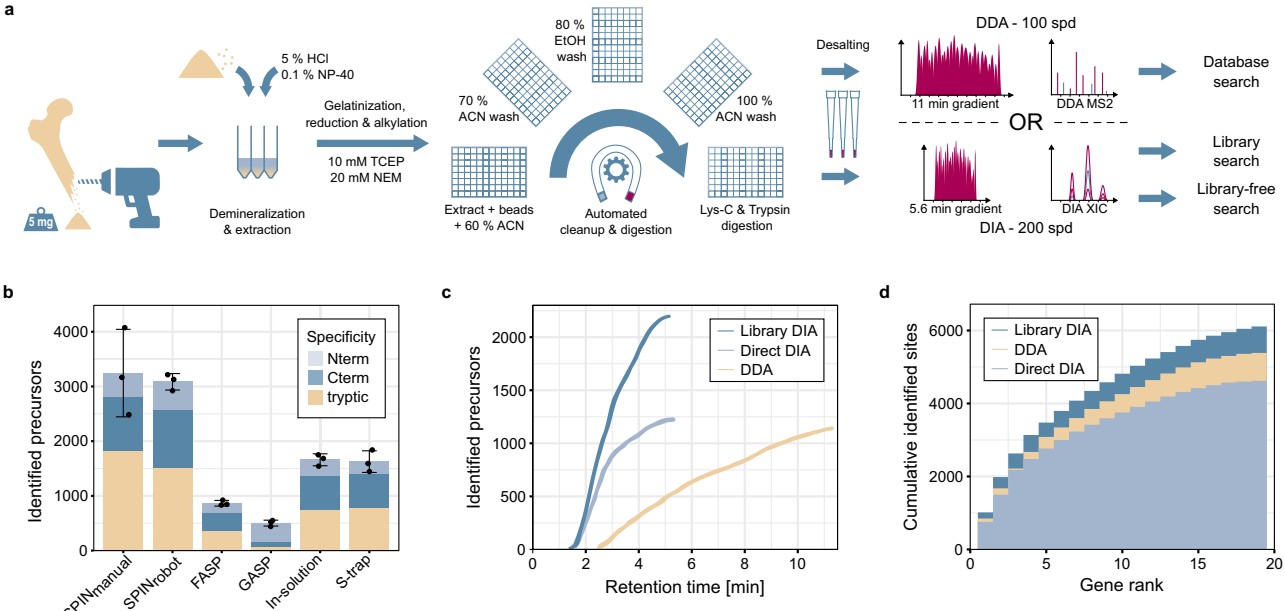

**Fig. 1 High-throughput bone proteome analysis workflow and benchmark. a** Sample preparation and data acquisition. Proteins were obtained from bone chips or powder by simultaneous demineralization and extraction. Cleanup by Protein Aggregation Capture and digestion can be automated by a magnetic bead-handling robot. Peptides were rapidly separated and analyzed by tandem mass spectrometry with a throughput of 100 samples per day (spd) in data-dependent or 200 spd in data-independent acquisition mode. **b** Performance of the "Species by Proteome INvestigation" (SPIN) sample preparation protocol executed manually (SPIN_manual) or by robot (SPIN_robot), compared to other common sample preparation techniques. Methods were compared by analyzing Pleistocene mammoth bone powder from the same batch in $n = 3$ technical replicate experiments with a 60 spd gradient and data-dependent acquisition. Bars indicate mean precursor identifications obtained with each method separated by enzymatic cleavage specificity shown by color with sand color for tryptic, dark blue for semi-tryptic with non-tryptic C-terminus and light blue for semi-tryptic with non-tryptic N-terminus. Error bars are centered at the total number of precursor identifications and indicate standard deviation. **c** Comparison of precursor identifications accumulated over retention time between the fast 100 spd DDA method analyzed by conventional database searching (sand color) and the rapid 200 spd DIA method analyzed with a library-based (dark blue) vs. library-free (light blue) approach. Peptides were generated by SPIN using a bovine bone and analyzed by LC-MS/MS with the two acquisition methods. **d** Gene-wise cumulative absolute amino acid coverage based on the precursors identified in **c** shown over the top 20 genes ranked by the number of precursors. Color indicates acquisition and peptide identification method with sand color for DDA, dark blue for library-based DIA, and light blue for library-free DirectDIA.

protocol consisting of only a few manual steps for easy scale-up (Fig. 1a). A combination of hydrochloric acid and the non-ionic detergent NP-40 in the extraction buffer facilitated demineralization and protein extraction in the same mixture, which otherwise usually requires intermediate centrifugation and buffer exchange[35]. Compared to other tested combinations, HCl and NP-40 were the only candidates that enabled efficient protein cleanup without precipitation (Table S1) and high peptide identification rates in the subsequent LC-MS/MS analysis (Fig. S1). We optimized the demineralization (Fig. S2) and extraction times (Fig. S3) and tested the effect of reduction and alkylation during the extraction step (Fig. S4). Contaminants, detergents, and minerals were removed using a modified protein aggregation capture (PAC) protocol with optimized solvent amounts (Fig. S5), which was automated on a magnetic bead-handling Kingfisher robot starting with only 5 mg of bone material. When parallelized with fast analysis, the automated sample preparation enables a single laboratory operator to continuously process and analyze 200 bone samples per day (Fig. S17). We benchmarked our new sample preparation workflow against the commonly used "in-solution" digestion protocol[25], "filter-aided sample preparation" (FASP)[36], "gel-aided sample preparation" (GASP)[37], and the more recent "S-trap"[38] using 5 mg per replicate of the same Pleistocene Mammoth bone sample[39] for all methods (Fig. 1b). The number of identified peptide precursors by LC-MS/MS was the lowest for FASP and GASP, which was probably due to losses in the filter or gel. "In-solution" and "S-trap" performed about two-fold better,

although "in-solution" had a relatively poor digestion efficiency leading to more missed tryptic cleavages (Fig. S6). The protocol developed for SPIN produced significantly more peptide precursor identifications by almost a factor of two compared to S-trap and in-solution. When comparing robotic processing to manual sample preparation following the SPIN protocol, we observed similar numbers of peptide identifications, but better reproducibility with the robot. Importantly, from a practical standpoint, the sample preparation procedure in SPIN requires much less hands-on time than FASP and GASP, which cannot easily be scaled to 96-well format. Moreover, although the "in-solution" digestion protocol is relatively fast, the lack of protein cleanup makes it more susceptible to contamination problems, which complicates scale-up. Finally, from an economic standpoint, "S-trap" can be costly due to the requirement for proprietary filter devices, whereas the modified PAC workflow in SPIN can be performed with any type of magnetic beads and a simple magnet rack instead of a Kingfisher robot.

**Rapid peptide identification.** To maximize scalability and throughput, SPIN uses very short online LC gradients to speed up the chromatographic separation of the peptide mixtures in line with Orbitrap tandem MS. We achieved sufficient proteome coverage for species discrimination (Fig. 1c) at a throughput of 100 samples per day with data-dependent acquisition (DDA, Fig. S7) and 200 samples per day with data-independent acquisition (DIA, Fig. S8, Table S2)[26]. Consequently, data acquisition in SPIN is

roughly 20 times faster than common practice in palaeoproteomics (<10 samples per day)[40] and 4 times faster than high-throughput plasma proteomics methods (60 samples per day)[12]. In a modern bovine bone sample, fast-scanning DDA identified about 1200 peptide precursors in 11 min, while DIA reached the same number of identifications in 5.6 min when analyzed without a spectral library by directDIA (Fig. 1c). Almost twice as many peptide precursors were identified by analyzing the same file using a dedicated spectral library that was generated once for each species by DDA analysis of offline fractionated peptide mixtures[41] (Figs. S9 and S10). However, as spectral library-based DIA yielded more overlapping peptide precursors, it did not result in a proportional gain in absolute sequence coverage (Fig. 1d). Across the 40 modern reference samples, the DDA method had a median coverage of 3678 amino acids and thereby outperformed the spectral library-free directDIA approach with 3226 amino acids (Fig. S11). The highest median coverage of 4480 amino acids was achieved with library-based DIA. As expected, sequence coverage was highest for the two most abundant bone proteins, COL1A1 and COL1A2. Since there was almost no additional coverage gained by including more protein sequences for the database search than the top 20 most abundant protein-coding genes, we decided to focus the SPIN analysis on only those 20 genes and thereby reduce noise and simplify the protein sequence database assembly and alignment (Supplementary Data 1).

**Automated species inference strategy**. The completeness and quality of protein sequence databases are vastly different between taxa (Fig. 2b). Missing genes, gaps, and stretches of incorrect amino acid sequences can introduce a bias towards well-annotated species when the proteomics-based taxonomic assignment is performed based on simple metrics like the number of identified peptides or protein groups. Therefore, we built a species inference algorithm encompassing site-based species-to-species comparisons (Fig. 2a). It is based on a gene-wise multiple sequence alignment (MSA) for all protein sequences across all available mammalian species for each of the 20 most highly expressed protein-coding genes in bone (Supplementary Data 1). Further manual refinement was needed to remove faulty sequence inserts or obvious prediction errors like frameshifts and the SPIN sequence inference algorithm was configured to automatically remove species (21 out of 177 in the current database) that lack more than 5 out of the 20 genes, because species with too many missing genes could not be reliably assigned. For all species identified in this study, sequences for all 20 genes were available for all taxa, except the white-tailed deer (19 genes), the European bison (15 genes), and the aurochs (15 genes). To confirm that the sequence information of the 20 genes is sufficient for resolving the taxonomy of all the 156 species in the database, we built a phylogenetic tree based on the refined sequences of the 20 most highly expressed bone proteins. Reassuringly, we observed that its topology matched the currently established phylogeny based on morphological and genomic data (Figs. 2b and S16). The protein sequence alignment was also the basis for creating a "site-specific difference matrix" by performing every possible species-to-species comparison. For all pairwise comparisons, this matrix only considers sites that are known for both species and have different amino acids. The aligned database was also used for mapping the LC-MS/MS-based peptide identifications to the correct genes and locations ("Mapping to alignment", Fig. 2a). Once mapped, the peptide data could be converted to site-level by splitting peptide identifications into amino acid identifications. In this format, the data can be used to score every species-to-species comparison in the site difference matrix. To best integrate the multiple MS metrics in the scoring scheme, peptide intensity, precursor count,

peptide count, and maximum score, were scaled and combined into a normalized joined score (J-Score). The J-Score does not necessarily reflect the actual amino acid probability for each site but assigns a higher weight to amino acids with better underlying data. The summed J-Scores were used for determining the winner of every species-to-species comparison in the site-specific difference matrix (Fig. 2c). The algorithm allows for a single or multiple indistinguishable species to win most comparisons depending on the sequence coverage and the number of closely related species in the protein sequence database. We added an optional "fine grouping" step using a manually curated list of marker peptides to keep the phylogenetic placement between closely related species consistent, even at low sequence coverage. To this point, a species was assigned to every sample, even including the blanks. The algorithm includes two mechanisms for controlling and minimizing the false-discovery rate (FDR), one to identify samples with too low signal, like blanks, and a second one to control for species that are not yet in the database. Samples with low peptide intensity were removed through quantitative comparison to the abundance of autolysis-derived protease peptides, which we treated like a spike-in standard. The threshold was automatically calibrated based on relative protease intensity in laboratory blanks. The second control mechanism was aimed at the identification of species with insufficient sequence coverage, as this would lead to unreliable classification. Therefore, we extended the database with an equally sized set of decoy species (randomly generated *chimera* species). The final results were then ranked by sequence coverage and a cutoff was applied to keep the number of decoy identifications below 1%. The comparison of site coverage and relative protease abundance demonstrated that most of the blanks along with the empty samples were successfully removed using the two thresholds and that the coverage decreases with sample age (Fig. 2d). To ensure global FDR control and for easy comparability between samples, all samples analyzed in the entire study were processed in the species inference pipeline, together. The species results were collected in a global result table (Supplementary Data 2) and the proteomics results converted to site-level were used to create a consensus of the identified sequences for each sample (Supplementary Data 3).

**Validating SPIN with bones from known species**. We optimized and assessed the performance of the different data acquisition and interpretation strategies using a set of 49 known reference bones from 13 species (Fig. 3a). We compared the sequence coverage (Fig. S11) and shared and unique peptide identifications (Fig. S12) between the three data types library-based DIA, directDIA, and DDA. All samples were placed in the correct genus using library-based DIA, whereas spectral library-free directDIA could not differentiate human from chimpanzee, and DDA was not able to exclude goats for one of the eight sheep samples. Interestingly, all three methods performed equally well, when it came to the placement of taxa within the families. Within bovines, the domestic cattle (*Bos taurus*) could be distinguished from European bison (*Bison bonasus*) but not from the aurochs (*Bos primigenius*). The European bison itself could not be discriminated against from American bison (*Bison bison*) and yak (*Bos mutus*) and in one case, from zebu (*Bos taurus indicus*) (Fig. 3b). The closely related goat (*Capra hircus*) and sheep (*Ovis aries*), were correctly identified in all DIA analyses and 10 out of 11 samples in DDA analysis. Within equines, domestic horse (*Equus ferus caballus*) was successfully discriminated from donkey (*Equus africanus asinus*), but not the Mongolian wild horse (*Equus ferus przewalskii*) (Fig. 3c). Fine-grouping was not actually required to distinguish goat and sheep or horse and donkey, but it made the *caballus*/*przewalskii* classification more uniform.

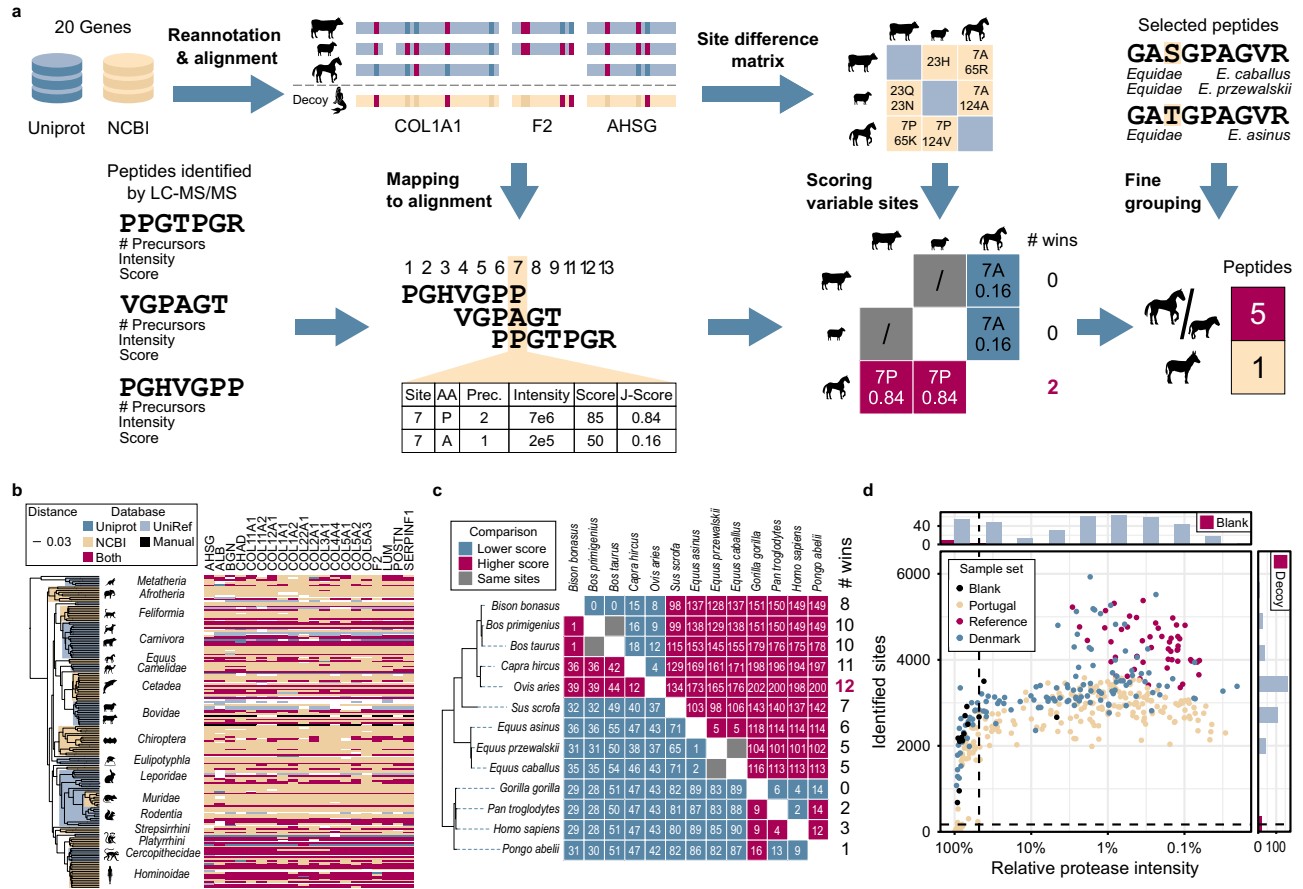

**Fig. 2 Data analysis pipeline for species identification. a** The aligned protein database, species difference matrix, and manually curated species marker peptides (top row) are used at multiple stages of the data processing pipeline (bottom row): Peptide identifications were converted to site-level, scored by joining intensity, score, and number of precursors (J-Score) and used to identify the winner of all possible species-to-species comparisons. Fine resolution of closely related species can be further improved by using manually selected species marker peptides. **b** The mammalian species database comprising 20 genes across 177 species (156 species with >14 genes) was generated by merging Uniprot and NCBI with manually curated and reannotated protein sequences. The phylogenetic tree was generated from the protein database using Fast Tree and FigTree. Color indicates database source with dark blue for sequences with annotated gene name from Uniprot, light blue for sequences from Uniprot with gene name added from UniRef, sand color for sequences from NCBI, pink for sequences available in both databases, and black for manually added sequences. **c** Example species competition matrix for the reference sample "Ovis_07" only showing the 13 reference species. White numbers indicate the summed joint scores (J-Score) of the species-discriminating sites. Gray cells indicate species pairs, where no species-discriminating sites have been identified in the sample. Pink indicates that the left species wins and blue indicates that the top species wins the comparison. The phylogenetic tree is a subset of the tree in panel **b**. The complete species competition matrices comprise all 156 target and 156 decoy species, i.e., 24,336 comparisons. **d** Absolute sequence coverage and relative protease intensity in reversed log10-scale for all samples from the three datasets in this study. The vertical site coverage cutoff is used to control the false-discovery rate at 1%. The horizontal protease intensity cutoff excludes samples with low signal (lower than 75% of the blank runs). Independent analysis of both parameters is displayed as histograms. Sample sets are indicated by color with black dots for blank runs, sand color for the Portuguese sample set, light blue for the samples from Denmark, and pink for the reference samples.

Besides common domesticated animals and their wild relatives, we explored the potential to detect and correctly identify great apes. While all three peptide identification methods could successfully classify human (*Homo sapiens*), orangutan (*Pongo abelii*), and gorilla (*Gorilla gorilla*), only DDA and library-based DIA analyses could correctly separate chimpanzee (*Pan troglodytes*) from *Homo sapiens* (Fig. 3d). It is noteworthy that both DDA and library-based DIA analysis of two chimpanzee bones assigned one of them to chimpanzee and the other to bonobo (*Pan paniscus*). Unfortunately, the low quality of the available bonobo protein database prevented closer investigation. These results confirmed that the SPIN workflow can be used to classify great apes at the genus level.

As a proof-of-concept, we investigated the potential to detect species hybrids with SPIN by analyzing two samples from mules. Focusing on two peptides that are distinct between horse and

donkey, one of the two mule samples showed high intensity for both sequence variants, as expected, but the second mule showed peptide intensities typical for a donkey (Fig. 3c). We concluded that SPIN is technically capable of hybrid detection, but an assessment of its reliability would require a larger study size.

**Performance and reproducibility benchmark.** To benchmark the SPIN analysis strategy against standard-practice bioarchaeological species determination based on bone morphology, we analyzed a set of 63 bone fragments related to human activities at the "Salpetermosen Syd 10" site (MNS50010, ZMK5/2013) in Denmark, which dates to the early Pre-Roman Iron-Age (380 BC–540 AD, Fig. 4a)[42]. Some of the bones showed strong signs of decay due to the age and the wet anoxic conditions in the Salpetermosen bog (Fig. 4b). Each specimen was morphologically

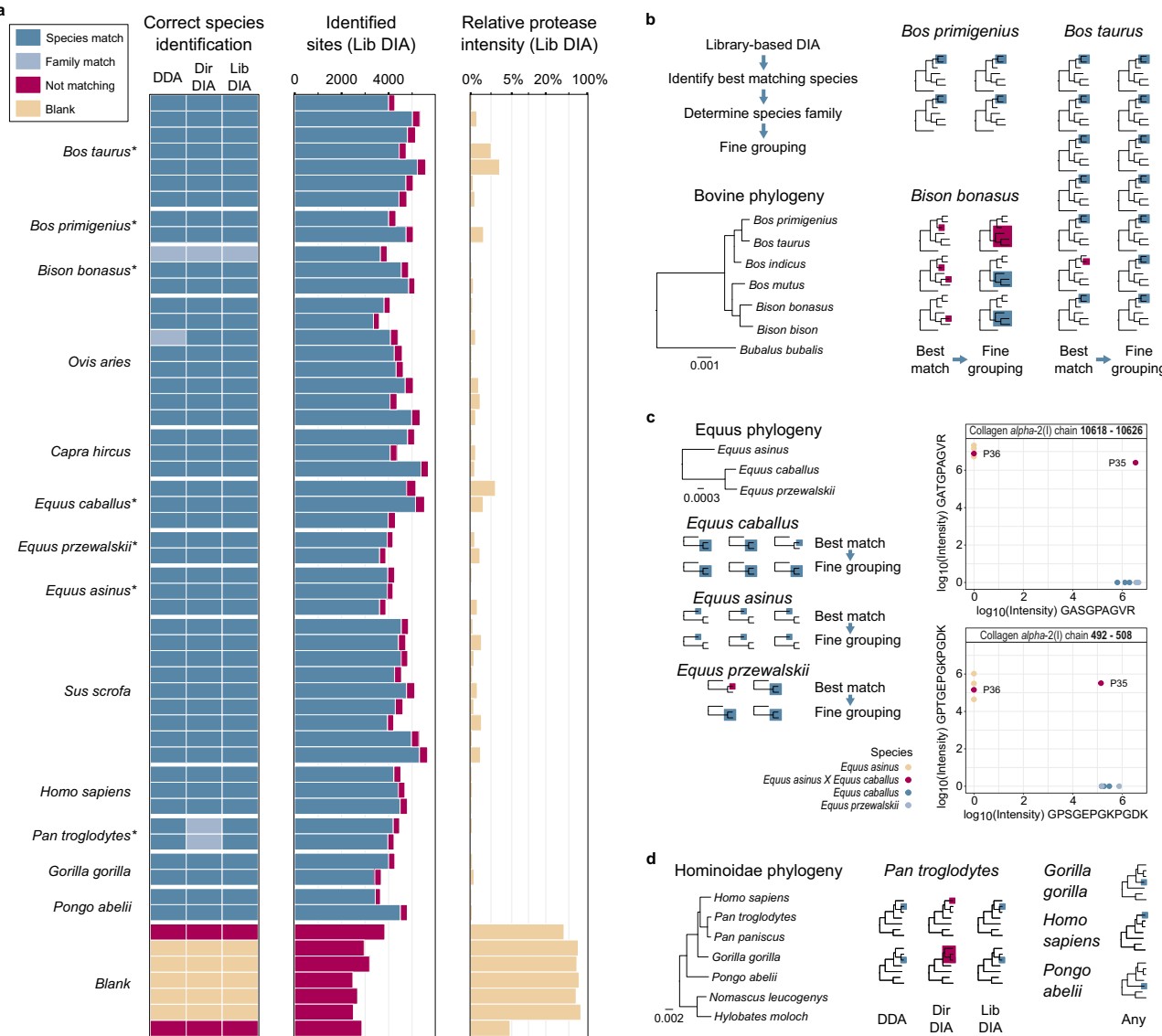

**Fig. 3 Reference species analysis. a** Species identification results based on DDA, directDIA, and library-based DIA analysis. Dark blue boxes indicate correct identification of a single or multiple indistinguishable (marked by asterisk) species. Light blue indicates species that could not be separated from their closest relatives. Blanks that were below the relative protease intensity threshold are shown in pink. The "identified sites" bar chart shows the absolute amino acid coverage in blue for sites matching the true species and pink for non-matching sites. The relative protease intensity was calculated by dividing the intensity of protease peptides by total intensity and plotted in log-scale. **b** Bovine species identifications obtained by library-based DIA analysis. Phylogeny was based on the protein database. Correctly identified single or indistinguishable species are highlighted in blue. Inconsistent identifications are marked in pink. Best Matching species are on the left and the refined "fine-grouping" on the right side. **c** Same display as in **b** for equine species analyzed by library-based DIA data. The additional plots on the right side show the log10 intensity of two species-discriminating peptides for the horse isoform on the x-axis and the donkey isoform on the y-axis. Missing quantifications are shown as zero log10 intensity. Donkey samples are marked in sand color, horse in dark blue, Przewalskii horse in light blue, and hybrids pink. **d** Species identifications after fine-grouping for great apes comparing the three different peptide identification strategies. Correctly identified *genus* is highlighted in blue. Broader matches within the family are marked in pink. Differences between the three identification strategies were only observed for the genus *Pan*.

analyzed by an experienced zooarchaeologist, and the SPIN analysis was conducted in technical duplicates starting on bone powder level. Variations in the input amount, peptide recoveries, and LC-MS/MS performance resulted in one experiment with higher and one with lower average MS intensity. The study was blinded by keeping the morphological species identification undisclosed until the SPIN analysis was finalized.

SPIN analysis using the 5 min DIA method and fine-grouping resulted in 49 exact and 3 approximate species identifications in the higher intensity experiment. In case of the lower-intensity replica experiment, we obtained 44 exact and 2 false species

identifications (Fig. 4c). The remaining 11 (high-intensity replicate) and 17 (low-intensity replicate) samples were excluded by the algorithm due to high relative protease intensity. The laboratory blanks were also correctly excluded. The comparison of replicates showed perfect reproducibility between duplicates with site coverage >3000 amino acids. We observed four cases of missing identifications in the high-intensity replicate that were identified in the low-intensity experiment, which we attributed to variability in the bone chips, input amounts, and peptide recovery. Importantly, there were no contradicting species identifications between the two replicates. For 94 out of 98

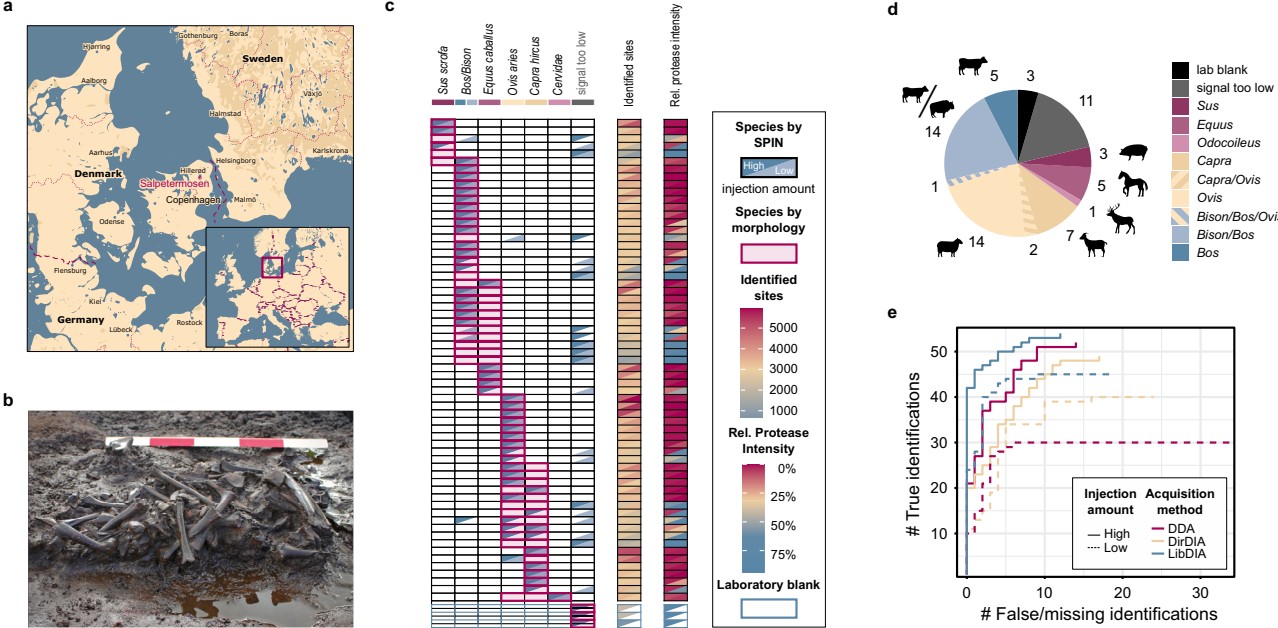

**Fig. 4 Species identification of bones from the Scandinavian Iron-Age. a** Location of the archaeological site "Salpetermosen Syd 10" on Zealand in Denmark in the Hillerød municipality 30 km north of Copenhagen. Map drawn in Mapbox Studio using a custom style. **b** Cross section of an in situ wetland bone deposit. Scale bar is 50 cm. Four bones were radiocarbon dated between 1720 and 1570 BP. Picture provided by the Museum of North Zealand. **c** Species identification results by SPIN (5 min method, library-based DIA) and by morphological assessment for 63 samples from the Salpetermosen site measured in technical duplicates and 3 blanks. Rows represent individual samples and have been ordered first by morphological species assignment and then by decreasing mean site coverage. The upper left and lower right wedge of each cell represent results measured in two separate experiments, one with higher (upper left, dark blue) and the other with lower (lower right, light blue) MS signal intensity. The first seven columns indicate SPIN species by blue wedges and morphological species possibilities by pink boxes. Bovine species assignments are combined in column two. The eighth and ninth columns are heatmaps showing the absolute number of covered amino acids and relative protease intensity, respectively. **d** Summary of SPIN species identifications from panel c in the replicate with high MS intensity. Bovine identifications are separated into cow (*Bos*) and broader bovine identifications (*Bos/Bison*). Striped colors indicate samples with insufficient sequence coverage to distinguish closely related taxa. Samples with insufficient sequence coverage for confident species identification are marked as "signal too low" and correctly excluded blanks are marked in black. **e** Pseudo receiver operating characteristic (ROC) curves for comparing the sensitivity and success rate of three different data acquisition and analysis strategies. Results of each dataset were sorted by decreasing number of identified sites. The *y*-axis shows the cumulative number of correct species identifications in agreement with the morphology. The *x*-axis shows the cumulative number of false or missing identifications below the relative protease intensity threshold. Color indicates data acquisition and analysis mode with pink for DDA, dark blue for library-based DIA, and sand color for library-free DirectDIA. Experiments with lower MS intensity are shown by dashed and high intensity by solid lines.

identified samples from both replicates, the SPIN analysis was in agreement with the morphological analysis (Fig. 4c). Two of the inconsistencies were sheep identifications for a sample that had the appearance of a goat bone, while the other two had low peptide intensity and were classified as cattle and sheep by SPIN but morphologically closer to pig and cattle, respectively. Sheep and goat, which often cannot be discriminated morphologically, were unambiguously identified by SPIN in 39 cases and could not be distinguished in two. For cattle, only *Bos* is plausible at this time and location[43] and the SPIN and morphological identifications showed good agreement. Nevertheless, we were interested in the performance of SPIN for distinguishing *Bos* and *Bison* in degraded material and therefore looked at the overall species distribution in the high-intensity replica experiment (Fig. 4d). Discriminating between *Bos* and *Bison* was only possible for 5 out of 14 bovine bones and became significantly more challenging with lower sequence coverage (Supplementary Data 2). Cattle and horses could not be distinguished morphologically in 10 cases, 7 of which could be resolved by SPIN. All three laboratory blanks were correctly excluded ("signal too low") by the relative protease intensity threshold.

We compared the performance of the three different types of peptide identifications by library DIA, directDIA, and DDA, which performed very similarly to the reference samples.

Compared to the well-preserved reference samples, the differences became much more apparent in the more degraded Salpetermosen sample set. The pseudo ROC-curve analysis shows that the DIA-based methods outcompeted DDA, especially in the low amount replicate, indicating higher sensitivity in DIA-based measurements (Fig. 4e). Between the two DIA methods, library-based DIA consistently produced more true species identifications than directDIA.

**Species identification of Middle and Upper Palaeolithic archaeological bones.** To challenge the SPIN workflow with highly degraded samples and demonstrate its scalability, we analyzed a set of 213 archaeological bone fragments from three Portuguese archaeological sites with early human occupation (Fig. 5a). To this end, we translated the output of the species inference algorithm to reflect the most likely ancestors that were present at the location and time (Table S3). Analogous to all other samples in this study, we analyzed the Portuguese bones with DDA and DIA, which took 52 h and 26 h of MS acquisition time, respectively. We used the library-based DIA results as the basis for species identification because of its higher resolution, as demonstrated with the Salpetermosen dataset. However, to allow the identification of species for which no spectral library is

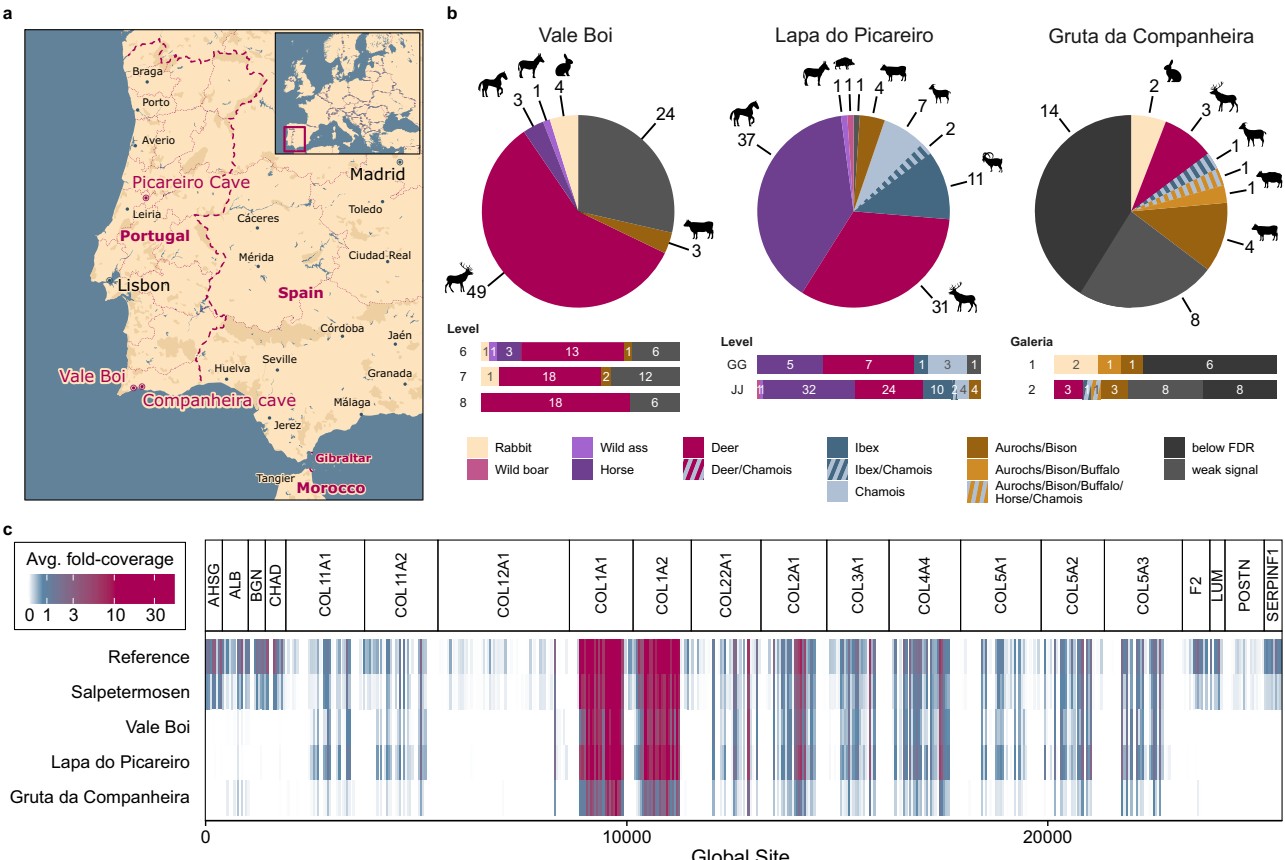

**Fig. 5 Large-scale species identification at three sites with early human occupation on the Iberian peninsula. a** Locations of the three sites on a current map of Portugal. Map drawn in Mapbox Studio using a custom style. **b** Species identified in 84 samples from levels 6–7 (29–31,500 BP) of Vale Boi, in 95 samples from layers GG to JJ (38–45,000 BP) of Lapa do Picareiro, and 34 samples from chambers 1 and 2 (estimated 50–60,000 BP) of Gruta da Companheira. Overall species distribution is displayed by the pie chart, whereas bar charts show species ratios for separate compartments of the assemblage. Colors are used to distinguish species, as indicated in the legend. **c** Average fold-coverage of the 20 genes used for SPIN comparing the three Portuguese sites with the modern reference and iron-age material. Coverage was calculated by summing the number of precursors at each site in the global aligned database and is indicated by color using white for no coverage, blue for medium coverage, and pink for high coverage. The values represent the average fold-coverage in 10 amino acid bins for each dataset.

currently available, such as rodents, we replaced the result with the taxonomy identified by directDIA, whenever directDIA detected such a species. As both results were based on the same raw data, the relative protease threshold remained unaffected, but the sequence coverage was lower with directDIA. In addition, compared to the reference and Salpetermosen samples, these Southern European Middle and Upper Palaeolithic samples suffered from reduced protein sequence coverage across the proteome assembly (Figs. 5c, S13) and an increase in protein deamidation (Fig. S14).

For Lapa do Picareiro, 94 out of 95 samples could be confidently assigned with a species. For these specimens, which were dated approximately between 38,000–41,000 BP (layers GG-II) and 45,000 BP (layer JJ)[44,45], species composition was relatively similar for both layers (Fig. 5b). Of particular interest was the identification of one specimen of the now-extinct European wild ass (*E. hemionus hydruntinus*), alongside 37 caballine horses. Most of the ibex and chamois bones, which are not easy to distinguish morphologically, could be uniquely assigned to one of the two (18 out of 20). Finally, bovines and wild boar were exclusively identified in the older "JJ" layer.

For Vale Boi, dated between 31,500 and 29,000 BP[46,47], 60 out of 84 samples could be confidently identified. The remaining 24 samples failed to meet the abundance-based quality threshold and were therefore not assigned a species identity (Fig. 5b). The

vast majority of the identified bones from layers 6 and 7 and all bones in layer 8 were classified as deer, which is in agreement with the previously reported numbers for large mammals[46]. Equids, including one *E. hydruntinus*, were only identified in layer 6. With directDIA, four smaller bone fragments from layers 6 and 7 could be classified as rabbits, which were highly abundant at Vale Boi[48].

Finally, for Gruta da Companheira 12 out of 34 samples could be assigned a confident species identification. Expected to date around 50,000–60,000 BP, 14 samples failed to meet the FDR threshold while eight samples were excluded due to failing to meet the abundance-based quality threshold. Amongst the confidently identified species at Gruta da Companheira were bovines, deer, and rabbits, whereas the only *ovicaprine* sample could not be uniquely assigned to either ibex or chamois. Although below the relative protease cutoff, the two most interesting bones, which were both found in *Galeria 2*, matched best to great apes. The sample with the highest sequence coverage (1817 aa) was classified as human or chimpanzee (Fig. S15), whereas the sample with lower coverage (235 aa) matched equally well with all great apes. Here, the SPIN results can be used as a starting point for future in-depth protein and ancient DNA analyses to find out whether these are actually human remains and to eventually define their genetic profile[40].

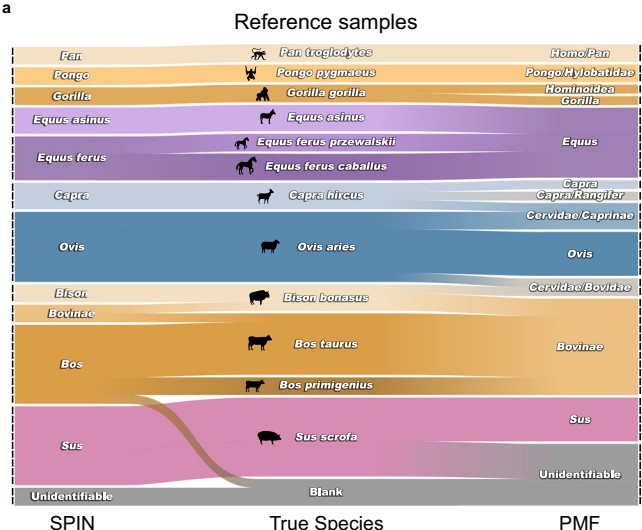

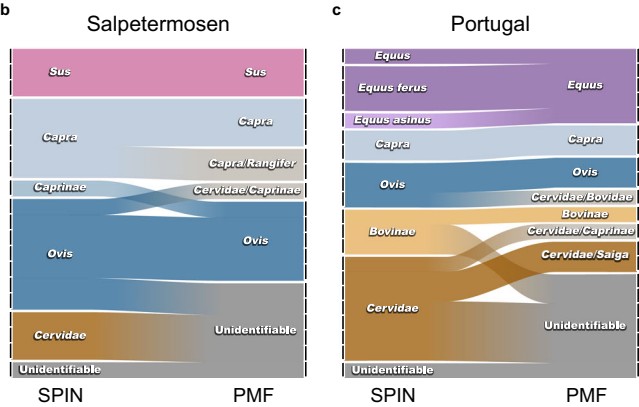

**Fig. 6 Comparison of SPIN and PMF. a** Alluvial diagram showing species identification of 46 reference bone samples and 3 laboratory blanks. Small bars on the *x*-axis indicate individual samples. Color and position in the middle column represent the true species, whereas the left and right column report the species identification by SPIN and PMF, respectively. Bars with color gradients indicate changing species assignments. **b** Alluvial diagram showing species identification of 20 representative samples from the Danish Salpetermosen site (Fig. 5). Left column indicates the species identification by SPIN, whereas the right column indicates the species identified by PMF. **c** Alluvial diagram showing species identification of 21 representative samples from the three Portuguese sites (Fig. 5). Left column indicates the species identification by SPIN, whereas the right column indicates the species identified by PMF.

**Performance comparison against collagen type I PMF.** To evaluate the SPIN method against the current method of choice for species identification by mass spectrometry, we compared it directly with MALDI-TOF MS-based species analysis of collagen type I PMF, i.e., ZooMS analysis[18,49]. To ensure a fair comparison, we assembled a representative test set of peptides comprising all 46 samples from our reference set, 20 from Salpetermosen, and 21 from the Portuguese bone assemblages (total = 87 samples), as well as 3 extraction blanks. SPIN and ZooMS analysis were performed using the same amount of peptides per replicate that were generated following the protocol developed for SPIN (Methods, Fig. 1a).

Amongst the known references, SPIN and ZooMS obtained the same level of taxonomic identity in 14 cases (30%), SPIN was more specific in 28 cases (60%), and ZooMS could not determine the species of 4 samples (9%), all of which were pig bones readily

identified by SPIN (Fig. 6a). In case of the material from Salpetermosen (Fig. 6b), the same level of identification was achieved for 10 samples (50%), more specific classification was achieved with SPIN in 3 cases (15%) and with ZooMS in 1 case (5%). Samples were not identifiable by SPIN in 1 case (5%) and by ZooMS in 6 cases (30%). For the most degraded material from the Portuguese sites (Fig. 6c), species identifications were on the same level for 10 samples (48%), more specific with SPIN for 4 samples (19%), and exclusively identified by SPIN in 6 cases (29%). We found no case in any of the samples, where the ZooMS and SPIN identifications were mutually exclusive. With ZooMS, all three laboratory blanks were all marked as "unidentifiable", whereas SPIN assigned one of them as *Bos taurus* (Fig. 6a). The majority of the ZooMS identifications (57 out of 87, 63.3%) provided a level of taxonomic specificity that cannot be improved further without adding more peptide markers beyond the nine we looked for. We therefore conclude that SPIN provides a level of taxonomic specificity unreachable by current ZooMS collagen PMF approaches, most likely due to the analysis of more divergent non-collagenous proteins, higher dynamic range achieved by chromatography, confidence control for peptide identifications, and higher resolution of the used MS instrument used for SPIN.

## Discussion

Here, we present a streamlined proteomics approach for genetic evidence-based species identification that bridges the gap between high-throughput low-cost targeted methods, such as PMF, PCR, or ELISA, and more powerful low-throughput high-cost approaches like NGS or conventional proteomics analysis by LC-MS/MS. We demonstrate that the automated sample preparation workflow based on PAC produces high-quality peptide samples for bone proteome analysis by LC-MS/MS and allows for easy scale-up. Data acquisition, which used to be a bottleneck in terms of speed and costs, was drastically shortened to reach a throughput of up to 200 samples per day per MS instrument. This became possible with a gradient storage-based LC system and advanced software for DIA data interpretation, two emerging technologies that have previously been undescribed in bone proteome analysis. The data interpretation strategy used for SPIN facilitates an unbiased comparison of currently up to 156 species and provides high confidence due to FDR control on both peptide and species identification levels.

We concluded that the highest confidence and sensitivity could be achieved with the library-based DIA approach, based on the analysis of 46 references and 64 more degraded bones from Denmark. These results were validated internally with replication and externally with morphological species identification. Furthermore, we demonstrated the high throughput of this method by analyzing a set of 213 Palaeolithic bones from Portugal in little more than a single day of MS time. SPIN performed well during the analysis of degraded samples and still had a reasonable success rate for 50,000–60,000-year-old bone material from a Mediterranean climate. In contrast, samples in a similar state of degradation require considerable effort for ancient DNA analysis. We expect, with an optimized sample preparation protocol for very stable specimens like teeth, SPIN will become applicable for palaeoproteomics analysis of million-year-old material[40].

By comparing SPIN with PMF analysis of type I collagen using the popular ZooMS protocol, we demonstrated that SPIN outperformed the species resolution of PMF in most cases. It needs to be noted that the species selection in the test set was not aimed at broad coverage of mammalian taxa but rather at the more challenging task of discriminating a few common species from their close relatives. Since we analyzed the same peptide extracts for fair comparability of the MS and data analysis performance,

the use of a PAC-based sample preparation method potentially impacted the PMF sensitivity. Nevertheless, many of the PMF-based species discriminations could not even be improved in an ideal scenario because they are inherently limited by the restriction to type I collagen. While a fair cost estimate for the two methods is hardly possible due to variable machine and labor costs, we believe that the shorter analysis time and automated data interpretation with SPIN will at least drive the costs of the two approaches closer together. In our case and possibly for many other groups, a major bottleneck in the PMF workflow was the manual analysis of peptide markers which was necessary because no universally accepted taxonomic assignment procedure of MALDI-TOF MS spectra was available at the time of this study.

So far, the SPIN data analysis is limited to 156 species that can be analyzed with DirectDIA, providing good species separation, or alternatively 13 spectral libraries for library-based DIA analysis, achieving the best possible species identification. Depending on the research question and archaeological context, studies may require an extension of the protein database and the set of spectral libraries. Although genomes are available for a plethora of extant mammals[50], the main bottleneck towards a more comprehensive protein database in our eyes is the genome annotation and sequence prediction for understudied species. With the appropriate sequence database and reference samples at hand, generating spectral libraries is a relatively straightforward task if required for a project. We envision that the number of available reference data will grow with more research groups sharing their protein sequence databases and spectral libraries, in the future. Furthermore, the SPIN workflow itself will likely be improved and expanded over time. We think of it as a modular protocol that can serve as the foundation for "SPIN-off" methods with custom building blocks, like: (i) sample preparation modified to support protein extraction from heavily-processed food products, (ii) data acquisition adapted for different instruments, and (iii) data interpretation including sex identification. Furthermore, it can be adapted to resolve mixtures of proteins from multiple taxa or to quantify protein damage (Fig. S14), in the future. Finally, we anticipate that the SPIN workflow will make LC-MS/MS more accessible for everyone, due to the reduction of the analytical costs per sample and a high degree of automation.

## Methods

**Sample description.** Each bone sample in this study was taken with permission from the respective museum, curator, or institution and the impact was minimized by only removing necessary amounts. The list of all samples and museum identifiers can be found in Supplementary Data 4. A fragment of a Pleistocene mammoth bone from permafrost dated to ~43,000 BP[39,51] was used for optimizing methods. Reference samples for *Bos taurus*, *Ovis aries*, *Sus scrofa* (mandibles), and *Equus caballus* (phalanx) were from the mixed viking-medieval deposits of the archaeological site Hotel Skandinavien (Århus Søndervold) (ZMK139/1964) in Århus, Denmark. The Laboratory of Biological Anthropology, Department of Forensic Medicine, at the University of Copenhagen provided the human reference sample, dentine from a previously described[40] 200–400-year-old premolar from "Almindelig Hospitals cemetery on Østerbrogade" in Copenhagen, Denmark. Reference samples for *Bos primigenius*, *Bison bonasus*, *Capra hircus*, *Equus asinus*, *Equus primigenius*, mule (*Equus caballus* X *Equus asinus*), *Pongo pygmaeus*, *Gorilla gorilla*, and *Pan troglodytes* were provided by the Natural History Museum of Denmark. The collection of 63 bone fragments dated to the Danish Early Iron-Age was sourced from the "Salpetermosen Syd 10" (MNS50010) site in Denmark[52]. The set of 213 upper-palaeolithic bone fragments from three Portuguese sites consisted of 84 samples from level 6 (29,300 BP[46]), level 7 (30,400 BP[47]), and level 8 (31,500 BP, unpublished) from Vale Boi, 95 samples from layers GG-II (38.000–41,000 BP) and JJ (45,000 BP[44,45]) from Lapa do Picareiro, and 34 samples from Galeria 1 and 2 from Gruta da Companheira[53].

**Sampling.** The reference samples for *Bos taurus*, *Ovis aries*, *Sus scrofa*, *Equus caballus*, and *Homo sapiens* and the 213 Portuguese samples were sampled in a clean laboratory designed for ancient DNA and protein work at the Globe Institute, at the University of Copenhagen. The remaining samples were processed in a laboratory with measures against human protein contamination. Working areas and tools were decontaminated with 5% bleach and 70% ethanol, between samples.

Reference bones intended for the generation of spectral libraries were surface-decontaminated by mechanical ablation and small samples of <100 mg were removed using a rotary cutting tool, followed by crushing of the pieces by mortar and pestle. For high-throughput analysis, ~5 mg samples were collected by scraping the fracture site of bone fragments with a small chisel or pliers and transferred into a 96-well plate. At least three laboratory blanks, i.e., empty wells, were included for each project with at least 1 blank on every plate.

**Combined demineralization and extraction for SPIN.** Five milligrams of bone powder or chips were suspended in 100 μL of demineralization/extraction solution containing 5% HCl and 0.1% NP-40 (ThermoFisher Scientific, 28324) in ultrapure water. Demineralization takes place at room temperature (rt) with continuous shaking at 1000 rpm, for 16–24 h. Reduction, alkylation, and collagen gelatinization were facilitated by adding 10 μL 0.1 M tris(2-carboxyethyl)phosphine (TCEP, Sigma–Aldrich, C4706) and 0.2 M N-ethylmaleimide (NEM, Sigma–Aldrich, E3876) in 50% ethanol and 50% ultrapure water and shaking at 1000 rpm at 60 °C, for 1 h.

**Protein cleanup and digestion for SPIN.** The purification and digestion take place on a KingFisher™ Flex (ThermoFisher Scientific) magnetic bead-handling robot. Debris was removed from the protein extract by centrifuging the plate at $800 \times g$, for 5 min. Magnetic SiMAG-Sulfon beads (Chemicell, 1202) were washed and prepared at a final concentration of 5 mg/ml in 60% acetonitrile (ACN) and 40% water. In a deep-well KingFisher™ plate, 10 μL bead solution and 40 μL of the clear protein extract were briefly mixed. Protein aggregation capture (PAC) was initiated by the addition of 240 μL of 70% ACN and 30% water (60% final ACN concentration) and finalized by incubating with shaking at 800 rpm for 5 min and without shaking for 1 min. The robot was loaded with this plate, "wash I" (500 μL 70% acetonitrile, 30% water), "wash II" (500 μL 80% ethanol, 20% water), "wash III" (500 μL 100% acetonitrile, and the "on-bead-digestion" plate (100 μL 20 mM Tris pH 8.5, Sigma–Aldrich 10708976001, 1 μg/mL LysC, Wako 129-02541, 2 μg/mL Trypsin, Promega V5111). The programmed sequence was: (i) collect the beads at low speed for 3:30 min, (ii–iv) washes I-III with slow mixing for 2 min, (v) digestion at 37 °C with slow mixing for 1 h, and (vi) bead collection and removal. The digestion was finalized outside the robot with shaking at 800 rpm and 37 °C, overnight. The peptides were acidified with 10 μL of 10% trifluoroacetic acid (TFA, Sigma–Aldrich, T6508). One Evotip (Evosep, EV-2001) per sample was washed in ACN, soaked with isopropyl alcohol, and equilibrated with 0.1% TFA in water, according to the manufacturer's protocol. The equilibrated tips were loaded with 10 μL peptide solution and subsequently washed with 20 μL 0.1% TFA, before LC-MS/MS.

**Peptide fractionation for spectral libraries.** Peptides for spectral libraries were obtained either from $3 \times 5$ mg bone powder processed by robot-based SPIN or from 20 mg bone powder processed manually with the same workflow. The peptides were desalted using C18 (3 M Empore, 66883-U) StageTips[54]. After quantification by Nanodrop (Thermo Fisher Scientific) at $A_{280nm}$, 12 μg peptides were adjusted to pH 7–8 by adding one volume of 50 mM ammonium bicarbonate (ABC, Sigma–Aldrich, A6141). Offline fractionation by high-pH reversed-phase chromatography was carried out on an Ultimate 3000 HPLC (Thermo Fisher Scientific) operated with Chromeleon (6.8) and equipped with a 15 cm long, 1 mm i.d., 1.7 μm particle size C18 column (Waters ACQUITY Peptide CSH) and 5 mM ABC as buffer A and 100% ACN as buffer B. The gradient at a flow rate of 30 μL/min started at 6% B, was increased to 18% B over 55 min, to 25% B over 12 min, to 70% B over 3 min, followed by a column wash at 70% B for 7 min and re-equilibration at 6% B for 9 min. During the gradient, 12 fractions of equal size were collected. Blanks were run between the different species to reduce carryover. The fractions were acidified with 1% (final concentration) TFA and vacuum-concentrated. An equivalent of 250 ng peptides (25% of each fraction) was loaded on Evotips as described above.

**Protein extraction and digestion methods for comparison.** Each of the tested protocols "in-solution"[7], FASP[55], GASP[37], and S-trap[38] was conducted in triplicates using 5 mg bone powder from the Pleistocene mammoth bone test sample. Demineralization was done by adding 100 μL of 5% HCl and incubating at r.t. and shaking at 1000 rpm, for 24 h. Insolubles were separated by spinning the suspension at $5000 \times g$ for 5 min and the supernatant was discarded or kept for FASP. The protein pellet was washed with 100 μL ultrapure water followed by repeating the centrifugation and discarding the supernatant. Proteins were extracted either in 100 μL 3 M guanidinium hydrochloride (Gnd-HCl, Sigma–Aldrich, G3272) in 0.1 M Tris, pH 8.5 for "in-solution" and FASP or in 100 μL 2% SDS (Sigma–Aldrich, 428018) for GASP and S-trap. To all protein extractions, 10 mM TCEP and 20 mM chloroacetamide (CAA, Sigma–Aldrich, C0267) were added and the samples were incubated with shaking at 1000 rpm at 60 °C, for 1 h. The protein concentrations of the extracts were quantified by BCA assay (ThermoFisher Scientific, 23225).

For "in-solution", the guanidinium extract was incubated with 1:300 (protease$_{wt}$:protein$_{wt}$) LysC at 37 °C, for 1 h. Subsequently, the sample was diluted with 3 volumes of 25 mM Tris, pH 8.5 and 1:100 (protease$_{wt}$:protein$_{wt}$) trypsin was

added for overnight digestion at 37 °C. The protein digest was acidified with 1% (final concentration) TFA and desalted using StageTips.

For FASP, the guanidinium extract was spun at $20,000 \times g$, for 10 min, and the supernatant was mixed with 9 volumes 8 M urea in 0.1 M Tris, pH 8.5 and transferred and passed through a 2.5 kDA MWCO filter (Millipore, UFC500324) in 200 µL steps by spinning at $5000 \times g$ for about 10 min. The filter was washed with 200 µL 8 M urea in 0.1 M Tris, pH 8.5 the demineralization supernatant was mixed with 6 volumes 8 M urea in 0.1 M Tris, pH 8.5, and passed through the same filter in 200 µL steps. After the last wash with 200 µL 8 M urea in 0.1 M Tris, pH 8.5, 100 ng LysC in 100 µL 50 mM ABC was added to the filter. Predigestion took place at 37 °C with shaking at 1000 rpm, for 1 h. Next, 200 ng trypsin was added and the digestion continued at 37 °C, overnight. The peptides were collected by spinning the filter and washing the membrane with 100 µL Tris, pH 8.5, spinning, washing with 100 µL 40% ACN 60% water, and spinning for final collection. The flow-through was concentrated to <50 µL and acidified with 1% (final concentration) TFA, before desalting on C18 StageTips.

For GASP, 100 µL SDS extract was mixed with 100 µL acrylamide solution (37.5:1 Acrylamide:Bisacrylamide), for 20 min. Polymerization was started by adding 8 µL TEMED (Sigma–Aldrich, T9281) and 8 µL 10% ammonium persulfate (Sigma–Aldrich, A3678). The obtained gel was sliced into pieces and fixed in 50% methanol, 40% water, and 10% acetic acid, for 30 min. The supernatant was removed and the gel was sequentially washed with 1 mL ACN, 1 mL 6 M urea, 1 mL ACN, 1 mL ACN, 1 mL 6 M urea, 1 mL ACN, 1 mL 50 mM ABC, 1 mL ACN, and finally resuspended in 200 µL 50 mM ABC. For predigestion, 100 ng LysC was added and the sample incubated at 37 °C with shaking at 1000 rpm, for 1 h. Next, 200 ng trypsin was added and the digestion continued at 37 °C, overnight. The supernatant containing the peptides was collected and the gel pieces extracted using 200 µL ACN, followed by 200 µL 5% formic acid, and 200 µL ACN. The peptide solutions were combined, vacuum-concentrated, and desalted on C18 StageTips.

For S-trap, 1.2% (final concentration) phosphoric acid was added to the SDS extract. An equivalent of 10 µg extract was mixed with 1 volume 90% methanol, 100 mM triethylammonium bicarbonate, pH 7 (TEAB, Sigma–Aldrich, 18597) and loaded on an S-trap filter (Protifi) by spinning at $2000 \times g$, for 1 min. The flow-through was re-loaded three times, before washing the filter with 200 µL of 90% methanol, 100 mM TEAB, pH 7. For predigestion, 100 ng LysC in 100 µL 50 mM Tris, pH 8.5 was added and the samples incubated with shaking at 1000 rpm at 37 °C, for 1 h. Then, 200 ng Trypsin in 50 µL 50 mM Tris, pH 8.5 was added and the digestion continued, overnight. The peptides were collected by centrifugation at $2,000 \times g$, for 1 min, and sequential washing with 100 µL 50 mM Tris, pH 8.5, 100 µL 0.1% TFA, and 100 µL 0.1% TFA in 50% ACN. After concentrating the peptides by vacuum evaporation, they were desalted on C18 StageTips.

**LC-MS/MS data acquisition methods.** For SPIN by DDA, chromatography was carried out using the 100 samples per day (SPD) method of an Evosep One (Evosep, Odense, Denmark) operated with the Evosep plugin (1.4.381.0) in Chronos (4.9.2.0) and an analytical column made in-house using a laser-pulled 8 cm long 150 µm inner diameter capillary packed with 1.9 µm C18 particles (Reprosil, Dr. Maisch). Peptides were ionized by nano-electrospray at 2 kV and analyzed on an Orbitrap Exploris 480™ (Thermo Fisher Scientific, Bremen, Germany) MS operated with Xcalibur (3.1-4). Full scans ranging from 350 to 1400 $m/z$ were measured at 60 k resolution, 25 ms max. IT, 300% AGC target. The top 6 precursors were selected (30 s dynamic exclusion) for HCD fragmentation with an isolation window of 1.3 m/z and an NCE of 30. MS2 scans were acquired at 15 k resolution, 22 ms max. IT, and 200% AGC target.

DIA SPIN analysis was based on the 200 SPD method of an Evosep One and the same column, ESI, and MS instrument as for DDA. Full scans ranging from 350 to 1400 $m/z$ were measured at 120 k resolution, 45 ms max. IT, 300% AGC target. Precursors were selected for data-independent fragmentation in 15 windows ranging from 349.5 to 770.5 $m/z$ and 3 windows ranging from 769.5–977.5 $m/z$ with 1 $m/z$ overlap. HCD fragmentation was set to an NCE of 27 and MS2 scans were acquired at 30 k resolution, 45 ms max. IT, and 1000% AGC target.

Samples for method optimization and spectral libraries were measured in DDA mode. Method optimization experiments were analyzed on the 60 SPD Evosep One gradient and spectral libraries on the 200 SPD gradient using the same column, ESI, and MS instrument as described above. Full scans ranging from 350 to 1400 $m/z$ were measured at 60 k resolution, 25 ms max. IT, 300% AGC target. The top 12 precursors were selected (30 s dynamic exclusion) for HCD fragmentation with an isolation window of 1.3 m/z and an NCE of 30. MS2 scans were acquired at 15 k resolution, 22 ms max. IT, and 200% AGC target.

**Protein database for SPIN.** The database includes all protein sequences from UniProt knowledgebase[56] release 2020_06 and NCBI RefSeq[57] release 201 (July 2020) from mammalian species and matching to the top 20 genes (Fig. 2b) in.fasta format. The NCBI entries were reannotated with gene and species information using the respective GenPept files and the fasta-headers were changed to a pseudo-Uniprot format: ">NCBI | [protein ID] | [protein ID]_[gene alias] [protein description] OS = [species name] OX = [species ID] GN = [gene name]". Relevant Uniprot entries with missing or false gene annotations were added by sequence similarity-based reannotation. The UniRef90 (release 2020_06) repository was used

to annotate each "90 % similarity" cluster with its most common gene name, followed by downloading all additional proteins matching the top 20 genes and updating the fasta-headers to include the correct gene name. Protein sequences from species missing in the databases like *Equus przewalskii*, *Bison bonasus*, and the extinct *Bos primigenius* were manually extracted from the available genomes[58–60] using reference sequences of the closest living relatives and the local BLAST[61] and visualization in UGENE[62]. After combining all protein sequences, filtering for mammalian species, and removing duplicates, the sequences were split into 20 separate files for each gene.

A multiple sequence alignment (MSA) was performed for each file using MUSCLE version 3.8.425[63] and the alignment was visualized in AliView version 1.26[64]. Upon manual inspection, faulty sequences, for instance, large inserts that were not shared by any other species or frameshifts identified by very low similarity to the rest of the alignment, were removed or changed into gaps. The aligned and manually refined databases were combined into one.fasta file with gene-wise alignment and a second gapless file for use in search engines. Phylogenetic trees based on this database were made by merging all genes of each species in alphabetical order, generating a consensus, generating the tree with FastTree version 2.1.11[65], and visualization with FigTree version 1.4.4 (Andrew Rambaut).

**Peptide identification.** All raw files from DDA were analyzed in MaxQuant version 1.6.0.17[66]. Variable modifications included oxidation (M), deamidation (NQ), Gln -> pyro-Glu, Glu -> pyro-Glu, and proline hydroxylation, whereas NEM-derivatization of Cys was configured as fixed modification. Spectral libraries and method optimization runs were searched against all protein sequences in the UniProt knowledgebase for the given species (download dates available in the MaxQuant "summary.txt"). Searches were first run with tryptic specificity and up to 2 missed cleavages to reduce the database size and then searched again with semi-tryptic specificity allowing for a peptide length between 8–30 aa and max. mass of 4000 Da. SPIN files were searched against the abovementioned gapless database only with semi-tryptic specificity using the same settings. Up to 5 variable modifications were allowed in tryptic and up to 4 in semi-tryptic searches. In all searches, "Second peptides" search and "Match between runs" were disabled, the score thresholds for identification were set to a minimum Andromeda score of 40 and delta score of 6, and the internal MaxQuant contaminant list was replaced with a custom database (Supplementary File "PR200512_HumanCons.fasta"). All other settings were left as default.

All raw files from DIA were analyzed in Biognosys Spectronaut[67,68] version 14.5.200813, or version 14.11.210528 for the files analyzed in the SPIN vs. MALDI-TOF comparison, using either library-based or library-free DirectDIA search. Spectral libraries were imported from the individual semi-tryptic MaxQuant search results with default settings except for digestion specificity and merged into a single library, before searching. The library-based search was carried out with default settings, except for MS1 as "Quantity MS-Level". The DirectDIA search was configured with the abovementioned gapless and custom contaminant databases. Digestion specificity was set to semi-tryptic with a peptide length between 7–40 amino acids, up to 2 missed cleavages, and the variable modifications included oxidation (M), deamidation (NQ), Gln -> pyro-Glu, Glu -> pyro-Glu, and proline hydroxylation, whereas NEM-derivatization of Cys was configured as fixed modification. Again, the "Quantity MS-Level" was set to MS1 and all other settings were kept as default.

**Species inference.** The species determination based on peptides identified with library-based DIA, DirectDIA, or DDA was done in R version 4.0.3 using RStudio version 1.3.10.93 and additional packages (reporting summary). The required peptide identification data was either generated with a Spectronaut report based on the SPIN.rs scheme (Supplementary Files) for DIA or extracted from the Maxquant output "evidence.txt" for DDA. Additionally, the aligned protein sequence database, the contaminant database, the experimental annotations, a list of species in the spectral library, and the fine-grouping table are needed (Supplementary Code[69]). The workflow was almost identical for the three types of data with small differences between MaxQuant and Spectronaut output column names, which can be found in the provided R scripts[69].

Species inference starts with loading the precursor identification files and filtering for 1% FDR. The databases were loaded and all isoleucines in the database and precursor sequences were changed into leucines. To allow for unambiguous site assignment, "global sites" were determined for every amino acid in the database by putting the 20 aligned genes in alphabetic order and numbering the positions from 1 through 25,550. The protein database was extended with an equal amount of "species decoy" proteins, which were generated by slicing the globally aligned sequences into 500 amino acid-long pieces and combining slices from randomized species into a "chimeric" decoy species. The combined target/decoy database was used to annotate the precursors with matching genes, proteins, species, and global sites covered by the peptide sequence. Site-level information was generated by counting precursors/peptides, finding the maximum score, and summing the intensity for every possible amino acid at every possible global site, for each raw file. The summed intensities larger than 1 were $\log_{10}$-transformed and the precursor count, peptide count, log-intensity, and max. score were scaled by dividing by the maximum at the global site. These four metrics were then multiplied to calculate a joint score (J-score), which was again scaled by dividing by the maximum J-score at the global site.

Based on the combined target/decoy database, a site difference matrix was generated. For every possible comparison of two species, the difference matrix lists the global sites and amino acids that are different between them. Global sites with gaps in one of the species are ignored. For each raw file, these species-discriminating sites were then scored using the J-score. The species with the higher J-score sum was selected as the "winner", while both species are "winners" in case of a "tie". The best match for a raw file was the species that won the most comparisons.

Fine-grouping was done for raw files with the best match that was amongst the species with manually selected marker peptides. The fine-grouping uses a list of hand-picked peptides that can be used to discriminate closely related species within a genus. These peptides were scored based on their precursor intensities in the respective sample. Based on the highest score, a single or multiple indistinguishable fine-grouping species were reported.

The final output species was selected by taking either the best match or, whenever available, the fine-grouping species. The "site count", which is the absolute sequence coverage, was added to this output table by counting the number of matching sites that were identified in the respective raw file. After ranking the list of raw files by decreasing site count, a q-value can be calculated as the fraction of decoy-species identifications at a given site count cutoff. Files with a q-value above 1% were marked in the final output. Furthermore, the relative protease intensity was calculated for each sample by dividing the summed precursor intensity of proteases peptides by the total precursor intensity sum. The relative protease intensity cutoff was determined as the upper quartile of relative protease intensity amongst the laboratory blanks. Samples below that threshold will also be marked in the final output.

**MALDI-TOF MS spectral acquisition and PMF analysis.** For the comparison between PMF- and LC-MS/MS-based species identification, 90 samples and 3 extraction blanks (Supplementary Data 5) were analyzed with both methods. Peptides generated using the SPIN protocol for protein extraction and digestion were split into a 10 μL aliquot for LC-MS/MS and a 30 μL aliquot for triplicate MALDI-TOF analysis. For desalting, the 30 μL were loaded and washed on Evotips as described in "Protein cleanup and digestion for SPIN" and subsequently eluted with two times 20 μL 50% ACN. After vacuum concentration to dryness, the peptides were reconstituted in 3.5 μL 50% ACN and 1 μL was spotted on a MALDI target plate for each replicate and mixed with 1 μL of CHCA (α-cyano-4-hydroxycinnamic acid). MALDI-TOF MS data acquisition was performed using an AutoFlex LRF MALDI-TOF MS instrument (Bruker) at the Fraunhofer Institute (Leipzig, Germany) in reflector mode, positive polarity, matrix suppression up to 590 Da, and spectral collection in the range of 700–3500 $m/z$. Spectra were exported to.txt files, their baselines were removed, were aligned to their respective triplicate replicates, and subsequently merged through R (R Core Team, 2018). Masses of the nine common peptide markers observed for mammalian species were taken into account following the nomenclature of Brown[70] and the peptide marker database presented by Welker[71]. Instead of peptide marker COL1α1 586–618, which contains a tryptic cleavage site and was therefore cleaved during efficient PAC digestion, the fully tryptic peptide COL1α1 604–618 in the mass range 1281–1327 Da was used.

**Reporting summary.** Further information on research design is available in the Nature Research Reporting Summary linked to this article.

## Data availability

The raw mass spectrometry proteomics data generated in this study have been deposited to the ProteomeXchange Consortium via the PRIDE[72] partner repository under accession code PXD024487. The processed proteomics data are available as part of the same repository and accession code. The data interpretation results generated in this study are provided as Supplementary Data files.

## Code availability

Scripts in R written for usage in RStudio and the sequence-aligned protein database are publicly available under Creative Commons Attribution 4.0 International license on Zenodo (https://doi.org/10.5281/zenodo.6406044)[69].

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

## Acknowledgements

We would like to express our gratitude to Daniel Klingberg Johansson and the Natural History Museum of Denmark for providing samples from non-human primates, Eske Willerslev for providing the mammoth bone, and Pedro Horta and Cláudia Costa for supporting the sample selection. We thank Meaghan Emma Mackie and Matthew Collins for helping to explore the PAC methods for very ancient samples. We thank the IZI Fraunhofer (Leipzig, Germany), Stefan Kalkhof, and Johannes Schmidt for access to the MALDI-TOF MS instrument, and Dorothea Mylopotamitaki and Huan Xia for assistance with generating the MALDI-TOF MS spectra. Thanks to Ulises Hernández Guzmán for naming our method, and to Dorte Breinholdt Bekker-Jensen and Tanveer Singh-Baath for providing technical foundations for our workflow. P.L.R., E.C., and J.V.O. were supported by the European Commission through the MSC European Training Network 'TEMPERA' (grant number 722606). Work at The Novo Nordisk Foundation Center for Protein Research (CPR) is funded in part by a generous donation from the Novo Nordisk Foundation (NNF14CC0001). This work has been supported by EPIC-XS, project number 823839, funded by the Horizon 2020 programme of the European Union. E.C. has received funding from the European Research Council (ERC) under the European Union's Horizon 2020 research and innovation program (grant agreement No 101021361) and from VIL-LUM FONDEN (No. 17649). F.W. has received funding from the European Research Council (ERC) under the European Union's Horizon 2020 research and innovation program (grant agreement No 948365) and a Marie Skłodowska Curie Individual Fellowship (No. 795569). The work in Vale Boi and Companheira is funded by Fundação para a Ciência e Tecnologia (FCT), grant PTDC/HAR-ARQ/27833/2017. The work at Lapa do Picareiro is funded by U.S. National Science Foundation (NSF) awards to J.H. (BCS-1420299, BCS-1724997) and M.B. (BCS-1420453, BCS-1725015). L.F. was supported by an SGS grant of the University of West Bohemia (SGS-2022-002). J.C. is funded by Fundação para a Ciência e para a Tecnologia (FCT), contract references DL57/2016/CP1361/CT0026. R.M.G. is funded by the European Regional Development Fund (FEDER) via the Programa Operacional CRESC Algarve 2020, of Portugal2020 (project ALG-01-0145-FEDER-29680), and the Fundação para a Ciência e a Tecnologia (FCT; contract reference 2020.00499.CEECIND). We thank all members of the JVO group for the constructive feedback, the MS platform team at CPR for keeping our instruments in good shape, and Thermo Fisher Scientific in Bremen for early access to MS instruments.

## Author contributions

P.L.R., J.V.O., and E.C. conceived the study. E.C. and F.W. established collaborations. P.B., K.M.G., P.P., M.C., R.M.G., L.F., J.C., M.L.S.J., M.M.B., J.H., N.B., F.W., and E.C. contributed archaeological samples. P.L.R., I.H., F.W., and A.J.T. performed laboratory research. P.L.R., I.H., and F.W. analyzed the data. P.B., K.M.G., P.P., M.C., R.M.G., L.F., J.C., M.M.B., J.H., N.B., F.W., and E.C. provided archaeological interpretations. P.R. prepared the manuscript. J.V.O., F.W., and E.C., and all co-authors contributed to the revision of the manuscript.

## Competing interests

The authors declare no competing interests.

**Additional information**

