## [Peer Review File · Nature Communications]

SPIN enables high throughput species identification of archaeological bone by proteomicsReviewers' Comments:

Reviewer #1:

Remarks to the Author:

In this manuscript, R  ther et al. propose a new high-throughput method to identify mammalian archaeological bones with the ability to expand it to other species in the future. This is a major improvement over the current higher throughput species identification that is currently used. They use state of the art technology and a new search strategy to analyze their samples. The comprehensive supplement describing method optimization is helpful and adds to the overall approach. The downside of this approach is its reliance on state-of-the-art technology that is not available in general and is cost prohibitive for most researchers who may want to apply it. As a result, I'm not convinced that this costs less per sample (Line 283-285) than other methods. This SPIN method uses a very expensive mass spectrometer, a specialized relatively new LC system, a robot system, and a proprietary software package. I agree that this approach reduces costs from a high-throughput standpoint, but the required instrumentation makes it generally prohibitive. Additionally, library generation is quite expensive (even sharing generated libraries has a cost associated with the original generation).

One other major point, the figures in the main manuscript have too many parts and are unnecessarily combined. NatComm allows up to 10 figures, so splitting them to make them more clear is definitely required for publication (e.g., Figure 1b is unreadable and not understandable as a result). Also many of the figures in both the main text and supplement are not colorblind accessible and should be revised.

Specific comments:

Line 55: For citation of protein aggregation capture, please include Cleland 2018 to reflect the first usage of this technology on bone in paleoproteomic-type studies.

Cleland, T. P. (2018). "Human Bone Paleoproteomics Utilizing the Single-Pot, Solid-Phase-Enhanced Sample Preparation Method to Maximize Detected Proteins and Reduce Humics." *Journal of Proteome Research* 17(11): 3976-3983.

Line 92: I agree that FASP and GASP don't translate to 96 well format, but S-trap has existing 96 well format. Please revise.

Line 102-103: Somewhere in this section or above, please provide the total time from bone powder to completion of identification or completion of the LC run.

Line 114 & 467: What were the 20 chosen genes? How were they chosen? Please add a table somewhere to include them.

Line 271: Thermal age is not real. Please reword this to reflect that the environment of deposition impacts preservation regardless of age. These samples are actually 50-60 kDa with less than ideal preservation.

Line 324-339: Please mention here that the accession numbers and museum information is available in supplementary tables for clarity.

Line 353-354: Is the extraction sequential or together (i.e., NP-40 in HCl?). It is unclear what the procedure is based on the supplementary optimization. Demineralization is 16 hrs followed by 1 hr extraction at 80C, but if the NP-40 is there the whole time then extraction is also occurring. Please clarify.

Line 465-487: Thank you for providing the curated fasta files in the supplementary information.

Line 492: What complete database was used? It is unclear as written.

Line 500: What proteins are present in the custom contaminant database? Please provide a list in the supplement.

Supplemental Table S3/Results: Please clarify what protein capture means beyond A280 because it is unclear what methods had detectable protein by MS. As written, it looks like only HCl+NP-40, but clearly the MS results suggest something different.

Additionally, are the extraction agent concentrations a final concentration after being combined with the demineralization agent?

Fig S1b: 10 genes are listed, but the caption says five. Please revise.

Fig S2: This figure is quite misleading based on the color gradient. The gradient range is too small to be useful and it is quite difficult to differentiate the levels of deamidation for day 3 vs day 1. Also as constructed, day 2 looks to be considerably more deamidated than the other two days but is within 6% which may not be significant. Please revise this figure.

Fig S3: Like Fig S2, this coloration is misleading. Quantitative accuracy of deamidation level is likely greater than 0.4%, please revise this figure.

Fig S4/Reduction and alkylation: The y-axis scale is strange and makes this figure a little misleading. Can a split axis figure be generated instead?

Only a limited increase in identifications is present after reduction, but in acidic conditions disulfide bridges are prevented from reforming by protons as SH. This is a typical approach in top down proteomics (i.e., reduction and acidification but no alkylation). Was protonated cysteine included as a variable modification? It would be interesting to see if protonated cysteine increases the numbers of IDs.

Were treatments with NEM applied without reducing first for the noTCEP categories? Please clarify.

Protein aggregation supplemental: Please cite that phase separation of ACN in paleoproteomic bone samples was previously observed in Cleland 2018.

Fig S11: Please use colorblind friendly palettes. Please revise.

Protein deam by SPIN (supplemental): "Deamidation is a common authenticity marker for ancient proteins, as it correlates roughly with sample degradation and thermal age" As I mentioned in comments for the main manuscript thermal age isn't real. Schroeter and Cleland, 2016 suggested that deamidation is a preservational signal and the deamidation data below shows something similar based on the broad percentages (if that is what is shown in Fig S13; see next comment). After this paper was in review, Ramsøe et al. 2021 found the same thing in calculus samples and suggested caution in using deamidation for authentication. Please revise.

Schroeter, E. R. and T. P. Cleland (2016). "Glutamine deamidation: an indicator of antiquity, or preservational quality?" *Rapid Communications in Mass Spectrometry* 30: 251-255.

Ramsøe, A., et al. (2021). "Assessing the degradation of ancient milk proteins through site-specific deamidation patterns." *Scientific Reports* 11(1): 7795.

Fig S13: This figure is unnecessarily confusing. It is unclear what the axes represent. Are they deamidation percentage? Also the top left figure goes to 600%. Please split and revise this figure for clarity.

Supplement deamidation conclusions: "depends heavily on the preservation" Please cite Schroeter and Cleland 2016 here as well as above.

Fig. S15: Please use colorblind accessible colors. Red and green together are difficult to interpret.

Reviewer #2:

Remarks to the Author:

The authors present a presumably modified approach to using proteomic techniques at identifying species rapidly – looking at over 150 mammalian species within just over 7 minutes.

In general, the authors should be more cautious with claims that “molecular species determination is generally unexploited when it comes to studies with large sample size”, given that there have been several studies analysing hundreds or even thousands of specimens using MALDI.

As the authors points out, there are limitations to the MALDI approach but they seem unaware of the machine learning-based approaches to mass spectral interpretation and their criticism of the statistical control is a reference to identifying proteins via PMF, rather than species.

It seems unclear as to why the authors want to give a slightly modified version of a relatively standard proteomic workflow its own acronym – future readers of this work can simply cite Ruther et al. without such a need for a name – particularly just a slightly faster form of shotgun proteomics.

The general idea of using magnetic beads in the sample preparation has been advocated for bone samples by Cleland et al. and is therefore not novel.

The authors later clarify that the sample processing, even by robotic means, takes a day per specimen, whereas the mention of 7.2 minutes in the abstract, albeit does state ‘mass spectrometry analysis’ is unnecessarily misleading – this is mainly because your ‘SPIN’ workflow is not exclusively the 7.2 minutes mass spectrometry part, but the whole workflow presumably.

Overall this simply seems like a method comparison paper (e.g., PAC vs GASP vs FASP vs S-trap, etc.), and using a slightly faster gradient on the LC system. This aspect would suggest its suitability more for a proteomics journal, although the archaeological focus could make it more appropriate for an archaeological science journal.

More details regarding the actual advances in technology that this includes would improve the manuscript.

Also, it is unclear how the 13 spectra libraries relate to the 156 species.

Mention of ‘robot-based SPIN’ makes it unclear whether all SPIN is roboticised, which is what the reader is lead to infer – although later on in the manuscript, after reading the methods section, it becomes more clear that SPIN probably only relates to the extraction itself.

Unclear when samples should be sent for DDA or DIA.

Ultimately, this is a nicely presented method comparison paper for proteomics, but one lacking details relating to the actual method improvements, particularly those of technology and the advantages of choosing the use of a curated local database rather than the standard public ones (e.g., UniProt).

It is also unclear how much faster this is than normal LC-based approaches, but if this is only 7.2 minutes per sample, rather than for 150 samples (which I initially inferred) then this is not surprisingly fast, nor much of an improvement when compared to other approaches (which could get results in fewer seconds than this does in minutes).

Reviewer #3:

Remarks to the Author:

This study provides a fast workflow to identify over 150 mammalian species for archaeological bones. The experiment is highly automated and the analysis speed is significantly improved, which will promote the application of proteomics in archaeology. However, the authors underestimate the potential of the ZooMS method, which is also used to identify the species of bones. ZooMS with MALDI-TOF-MS could analyze more than 200 samples per day. ZooMS is also developing some automation search algorithm.

SPIN also uses some search against a complete protein database, how long is the search time in MaxQuant for one sample and the further process in DDA and/or DIA? If the number of analyzed sample per day and search time in SPIN are not superior to those of ZooMS, what's the advantage of SPIN? The authors should give more detailed comparison.

Whatever, I think the automated and fast protein extract and enzymolysis as well as MS/MS analysis procedure could be at least used to analyze complex protein mixtures of archaeological samples, such as food, adhesive, oil residues.

Point-by-point rebuttal to REVIEWERS' COMMENTS

Manuscript: Nature Communications (NCOMMS-21-11457-T) by Rütther et al.

The reviewers' comments are provided in **black** font.

Our new responses to the reviewers' comments are provided below in **blue** font.

Reviewers' comments:

Reviewer #1 (Remarks to the Author):

In this manuscript, Rütther et al. propose a new high-throughput method to identify mammalian archaeological bones with the ability to expand it to other species in the future. This is a major improvement over the current higher throughput species identification that is currently used. They use state of the art technology and a new search strategy to analyze their samples. The comprehensive supplement describing method optimization is helpful and adds to the overall approach.

We thank the reviewer for acknowledging the technical advances and for pointing out our comprehensive supplement in which all method optimizations are thoroughly described.

The downside of this approach is its reliance on state-of-the-art technology that is not available in general and is cost prohibitive for most researchers who may want to apply it. As a result, I'm not convinced that this costs less per sample (Line 283-285) than other methods.

In the manuscript, we do not claim to have lower analytical costs compared to *other methods*, as the costs are highly-dependent on local factors, such as labor costs and instrument availability. However, we *do* claim that our method has significantly lower costs than the conventional ~ 10 times slower LC-MS/MS methods.

This SPIN method uses a very expensive mass spectrometer, a specialized relatively new LC system, a robot system, and a proprietary software package. I agree that this approach reduces costs from a high-throughput standpoint, but the required instrumentation makes it generally prohibitive. Additionally, library generation is quite expensive (even sharing generated libraries has a cost associated with the original generation).

We agree with the reviewer that the instrument costs are initially high but more spread when using the method on a large scale as anticipated. However, we foresee that the enhanced protein cleanup (similar to Cleland 2018) and automation make this workflow usable at MS core facilities that provide significant cost savings. As described in the manuscript, the data can be analyzed without spectral libraries at slightly lower species resolution to avoid the costs associated with library generation. Moreover, the quadrupole Orbitrap instruments capable of this type of DIA analysis have been produced and sold in the thousands and are accessible at many proteomics core facilities around the world.

One other major point, the figures in the main manuscript have too many parts and are unnecessarily combined. NatComm allows up to 10 figures, so splitting them to make them more clear is definitely required for publication (e.g., Figure 1b is unreadable and not understandable as a result). Also many of the figures in both the main text and supplement are not colorblind accessible and should be revised.

We simplified figures 1b and 4c in the revised manuscript. We checked the colors in the main figures in greyscale and with simulated Deuteranopia and could not find

indistinguishable colors. Pie charts were additionally labeled with icons for better readability. Supplementary figures S9 and S15 were changed to another color palette for better readability with Deuteranopia. We will be happy to adjust additional figures, if the reviewer could point them out for us.

Specific comments:

Line 55: For citation of protein aggregation capture, please include Cleland 2018 to reflect the first usage of this technology on bone in paleoproteomic-type studies.

Cleland, T. P. (2018). "Human Bone Paleoproteomics Utilizing the Single-Pot, Solid-Phase-Enhanced Sample Preparation Method to Maximize Detected Proteins and Reduce Humics." *Journal of Proteome Research* 17(11): 3976-3983.

Added in the revised manuscript

Line 92: I agree that FASP and GASP don't translate to 96 well format, but S-trap has existing 96 well format. Please revise.

Thank you for bringing this to our attention. The sentence has been revised.

Line 102-103: Somewhere in this section or above, please provide the total time from bone powder to completion of identification or completion of the LC run.

We added a supplementary figure (Fig. S16) describing the approximate sample processing times and a guideline for the continuous analysis of 200 samples per day by a single laboratory technician.

Line 114 & 467: What were the 20 chosen genes? How were they chosen? Please add a table somewhere to include them.

We added supplementary table S7 that describes the genes included and excluded from the protein database. In line 467, we referenced figure 2b that contains the names of the 20 genes on top of the "heatmap".

Line 271: Thermal age is not real. Please reword this to reflect that the environment of deposition impacts preservation regardless of age. These samples are actually 50-60 kDa with less than ideal preservation.

Rephrased in the manuscript.

Line 324-339: Please mention here that the accession numbers and museum information is available in supplementary tables for clarity.

We have now added this information.

Line 353-354: Is the extraction sequential or together (i.e., NP-40 in HCl?). It is unclear what the procedure is based on the supplementary optimization. Demineralization is 16 hrs followed by 1 hr extraction at 80C, but if the NP-40 is there the whole time then extraction is also occurring. Please clarify.

Sorry for the confusion. NP-40 and HCl are indeed together in the same extraction buffer.

We have modified the sentence to clarify this : "... are suspended in X \$\mu\$ L of demineralization/extraction solution containing 5 % HCL and 0.1 % NP-40"

Line 465-487: Thank you for providing the curated fasta files in the supplementary information.

You're welcome :)

Line 492: What complete database was used? It is unclear as written.

Thanks for catching this. Revised the sentence.

Line 500: What proteins are present in the custom contaminant database? Please provide a list in the supplement.

The database is provided with the supplementary files. We have added the file name in the manuscript.

Supplemental Table S3/Results: Please clarify what protein capture means beyond A280 because it is unclear what methods had detectable protein by MS. As written, it looks like only HCl+NP-40, but clearly the MS results suggest something different.

Additionally, are the extraction agent concentrations a final concentration after being combined with the demineralization agent?

Revised the table description accordingly.

Fig S1b: 10 genes are listed, but the caption says five. Please revise.

Thanks for catching this. Revised the caption. Additionally, we revised the figure in a similar fashion as Fig. S2 and S3 to display the deamidation rate separately.

Fig S2: This figure is quite misleading based on the color gradient. The gradient range is too small to be useful and it is quite difficult to differentiate the levels of deamidation for day 3 vs day 1. Also as constructed, day 2 looks to be considerably more deamidated than the other two days but is within 6% which may not be significant. Please revise this figure.

Figure S2 is now split into a) and b) in order to separate identifications and deamidation rate.

Fig S3: Like Fig S2, this coloration is misleading. Quantitative accuracy of deamidation level is likely greater than 0.4%, please revise this figure.

Figure S3 is now split into a) and b) in order to separate identifications and deamidation rate.

Fig S4/Reduction and alkylation: The y-axis scale is strange and makes this figure a little misleading. Can a split axis figure be generated instead?

Only a limited increase in identifications is present after reduction, but in acidic conditions disulfide bridges are prevented from reforming by protons as SH. This is a typical approach in top down proteomics (i.e., reduction and acidification but no alkylation). Was protonated cysteine included as a variable modification? It would be interesting to see if protonated cysteine increases the numbers of IDs.

Were treatments with NEM applied without reducing first for the noTCEP categories? Please clarify.

Figure S4 is now split into a) and b) in order to separate cysteine-containing peptides from peptides without cysteine.

Protein aggregation supplemental: Please cite that phase separation of ACN in paleoproteomic bone samples was previously observed in Cleland 2018.

Added reference.

Fig S11: Please use colorblind friendly palettes. Please revise.

The colors in figure S11 serve a purely decorative purpose and the figure remains fully understandable in greyscale.

Protein deam by SPIN (supplemental): "Deamidation is a common authenticity marker for ancient proteins, as it correlates roughly with sample degradation and thermal age" As I mentioned in comments for the main manuscript thermal age isn't real. Schroeter and Cleland, 2016 suggested that deamidation is a preservational signal and the deamidation data below shows something similar based on the broad percentages (if that is what is shown in Fig S13; see next comment). After this paper was in review, Ramsøe et al. 2021 found the same thing in calculus samples and suggested caution in using deamidation for authentication. Please revise.

Replaced the term “thermal age” with “stage of degradation”. We agree with the reviewer and Ramsøe 2021 that deamidation needs to be treated as an authenticity with caution. Although the abundance of deamidated peptides should not be used for proving authenticity, their absence can possibly be used to disprove it. Hence, we believe that the reproducible quantification of peptide deamidation is a relevant feature of the SPIN workflow.

Schroeter, E. R. and T. P. Cleland (2016). "Glutamine deamidation: an indicator of antiquity, or preservational quality?" *Rapid Communications in Mass Spectrometry* 30: 251-255.
Ramsøe, A., et al. (2021). "Assessing the degradation of ancient milk proteins through site-specific deamidation patterns." *Scientific Reports* 11(1): 7795.

Fig S13: This figure is unnecessarily confusing. It is unclear what the axes represent. Are they deamidation percentage? Also the top left figure goes to 600%. Please split and revise this figure for clarity.

Simplified the figure, added labels to the axes, and explained the calculation of the deamidation rates in the caption.

Supplement deamidation conclusions: "depends heavily on the preservation" Please cite Schroeter and Cleland 2016 here as well as above.

Added the two above-mentioned citations.

Fig. S15: Please use colorblind accessible colors. Red and green together are difficult to interpret.

Color palette was adjusted to be colorblind accessible.

Reviewer #2 (Remarks to the Author):

The authors present a presumably modified approach to using proteomic techniques at identifying species rapidly – looking at over 150 mammalian species within just over 7 minutes.

We thank the reviewer for acknowledging the speed and capability of our workflow. However, we find it misleading that the reviewer doubts the originality of our protocol (“presumably modified”) without presenting a reference. To the best of our knowledge, there is currently no alternative fast proteomics-based species identification workflow, particularly not with automated sample preparation and data interpretation.

In general, the authors should be more cautious with claims that “molecular species determination is generally unexploited when it comes to studies with large sample size”, given that there have been several studies analysing hundreds or even thousands of specimens using MALDI.

We agree with the reviewer and removed this statement from the revised manuscript.

As the authors points out, there are limitations to the MALDI approach but they seem unaware of the machine learning-based approaches to mass spectral interpretation and their criticism of the statistical control is a reference to identifying proteins via PMF, rather than species.

For the revised manuscript, we conducted a comprehensive comparison between MALDI- and SPIN-based species identification. However, we could not test the machine learning-based approach (Gu & Buckley, 2018), as the authors were unable or unwilling to provide the software despite our repeated requests. We offer to share the email communication regarding the software availability with the editor if needed. Furthermore, we encourage the reviewer to provide references to alternative statistical approaches to validate species identifications by PMF, such that we can test them with our data.

It seems unclear as to why the authors want to give a slightly modified version of a relatively standard proteomic workflow

While the extent of modification is certainly subjective, we would like to challenge the reviewer, who questioned the originality of our approach without providing references. In our eyes, a “standard proteomics workflow” entails manual protein digestion using an “in-solution” or “filter-aided” (Ludwig et al., 2018) method, followed by LC-MS/MS analysis using data-dependent acquisition, and ending with peptide and protein identification. SPIN, on the other hand, combines non-standard sample preparation by protein aggregation capture, non-standard data-independent acquisition, and a novel automated data interpretation algorithm.

its own acronym – future readers of this work can simply cite Ruther et al. without such a need for a name –

Future readers are welcome to use the original or modified SPIN protocol, provided data, and scripts by citing our paper without using the name “SPIN”, as it is not a registered trademark. We and probably also the developers of “ZooMS” (Buckley et al., 2009) find that a short name greatly simplifies discussions about available methods.

particularly just a slightly faster form of shotgun proteomics.

It is misleading to claim that SPIN analysis is only “slightly faster” than conventional shotgun proteomics without providing a reference. We have provided multiple recent references using significantly (10 to 30 times) slower methods in the original and revised manuscript.

The general idea of using magnetic beads in the sample preparation has been advocated for bone samples by Cleland et al. and is therefore not novel.

We did not claim in the original manuscript that magnetic bead-based sample preparation was a novelty. The applicability has indeed been demonstrated by Cleland et al. and we added the missing reference in the revised manuscript. However, the method optimization (supplementary information) and automation targeted at bone protein extraction and cleanup are novel developments.

The authors later clarify that the sample processing, even by robotic means, takes a day per specimen, whereas the mention of 7.2 minutes in the abstract, albeit does state ‘mass spectrometry analysis’ is unnecessarily misleading – this is mainly because your ‘SPIN’ workflow is not exclusively the 7.2 minutes mass spectrometry part, but the whole workflow presumably.

The abstract has been revised to “at a throughput of 200 samples per day”. We added a supplementary figure (fig. S16) that illustrates how a single laboratory technician can operate the SPIN pipeline with a continuous throughput of 200 samples per day. The reviewer correctly recognized that the processing time per sample takes multiple days and that the high throughput operation requires parallelization.

Overall this simply seems like a method comparison paper (e.g., PAC vs GASP vs FASP vs S-trap, etc.), and using a slightly faster gradient on the LC system. This aspect would suggest

its suitability more for a proteomics journal, although the archaeological focus could make it more appropriate for an archaeological science journal.

Method comparison is only a minor aspect of this paper (fig. 1b) and we regret that the reviewer did not recognize the efforts that went into the development of the data interpretation algorithm (fig. 2), the validation (fig. 3, 4), and the large-scale application (fig. 4, 5) of the SPIN workflow. In our opinion, our work addresses key bottlenecks, such as speed and simplification, that prevented proteomics to be more accessible to a broader range of research applications.

More details regarding the actual advances in technology that this includes would improve the manuscript.

We revised the introduction to make the actual advances more clear.

Also, it is unclear how the 13 spectra libraries relate to the 156 species.

As described in the manuscript, the 13 spectral libraries have been generated based on experimental data, whereas the 156 species database was sourced from public protein sequence repositories (Uniprot, NCBI, see methods section). It would be helpful, if the reviewer could provide us with page and line numbers of paragraphs, where this is unclear.

Mention of 'robot-based SPIN' makes it unclear whether all SPIN is roboticised, which is what the reader is lead to infer

The robot-based workflow applied to all samples except method optimization runs, is illustrated in fig. 1a and continuously referenced throughout the manuscript. Samples that were prepared manually (fig. 1b, supplementary figures) have been labeled as "manual" in the original manuscript. We do not know how to make this clearer.

– although later on in the manuscript, after reading the methods section, it becomes more clear that SPIN probably only relates to the extraction itself.

Again, we regret that the reviewer did not recognize the efforts that went into other parts of this study, such as the development of the automated data interpretation algorithm (fig. 2), the validation (fig. 3, 4), and large-scale application (fig. 4, 5) of the SPIN workflow.

Unclear when samples should be sent for DDA or DIA.

Added an explanation and comparison of the three different data sources (see above).

Ultimately, this is a nicely presented method comparison paper for proteomics, but one lacking details relating to the actual method improvements, particularly those of technology and the advantages of choosing the use of a curated local database rather than the standard public ones (e.g., UniProt).

We hope that the revised introduction makes the method improvements clearer. We described the motivation for curating the protein sequences from the public UniProt and NCBI databases on page 3, lines 115 - 120. The procedure for generating the database were outlined on page 11, lines 457 - 471.

It is also unclear how much faster this is than normal LC-based approaches, but if this is only 7.2 minutes per sample, rather than for 150 samples (which I initially inferred) then this is not surprisingly fast, nor much of an improvement when compared to other approaches (which could get results in fewer seconds than this does in minutes).

This potential point of confusion has been addressed in the revised abstract. The analysis indeed takes a total of 7.2 min per sample and we do believe that this analysis time is exceptionally fast in the perspective of bone proteome analysis, where conventional LC-MS/MS acquisition methods are between 1.5 and 3 hours (Cleland, 2018; Tsutaya et al., 2019). In the added section covering the comparison of SPIN with PMF-based ZooMS, we discuss the differences in effective analysis time and demonstrate the superiority of SPIN

compared to ZooMS in respect to identifying the correct species across a multitude of mammalian bone samples of different origin.

Reviewer #3 (Remarks to the Author):

This study provides a fast workflow to identify over 150 mammalian species for archaeological bones. The experiment is highly automated and the analysis speed is significantly improved, which will promote the application of proteomics in archaeology.

We acknowledge the reviewer for pointing out the advances over conventional proteomics and highlighting that it will promote proteomics in archeology.

However, the authors underestimate the potential of the ZooMS method, which is also used to identify the species of bones. ZooMS with MALDI-TOF-MS could analyze more than 200 samples per day. ZooMS is also developing some automation search algorithm.

To replace estimation with evidence, we now added a direct comparison between SPIN and MALDI-TOF-based ZooMS analysis of the same sample set in the revised manuscript. It shows that SPIN achieves more precise species differentiation and in some cases better sensitivity. As mentioned in the replies to reviewer #2, the automated search algorithm for ZooMS could unfortunately not be tested due to it not being available to the community and the authors originally describing the algorithm did not respond positively to our repeated inquiries. We offer to share the email communication with the developers of the software regarding its availability with the editor if needed.

SPIN also uses some search against a complete protein database, how long is the search time in MaxQuant for one sample and the further process in DDA and/or DIA?

It is unfortunately hardly possible to provide a reliable search time for the MS data, as this heavily depends on the available computer hardware. We encourage the reviewers to reprocess the provided data to find the search time on their system.

If the number of analyzed sample per day and search time in SPIN are not superior to those of ZooMS, what's the advantage of SPIN? The authors should give more detailed comparison.

We agree and have now addressed this in the revised manuscript. In short, the main advantage of SPIN compared to the ZooMS protocol is better species resolution and automated data analysis at a comparable throughput.

Whatever, I think the automated and fast protein extract and enzymolysis as well as MS/MS analysis procedure could be at least used to analyze complex protein mixtures of archaeological samples, such as food, adhesive, oil residues.

We agree with the reviewer that these applications would be desirable future developments of our methods. Nevertheless, we want to emphasize that the identification of a single-species sample by proteomics is an analytical challenge that has received far less attention compared to complex protein mixture analysis, and therefore had to be solved.

References

- Buckley, M., Collins, M., Thomas-Oates, J., & Wilson, J. C. (2009). Species identification by analysis of bone collagen using matrix-assisted laser desorption/ionisation time-of-flight mass spectrometry. *Rapid Communications in Mass Spectrometry: An International Journal Devoted to the Rapid Dissemination of Up-to-the-Minute Research in Mass Spectrometry*, 23(23), 3843–3854.
- Cleland, T. P. (2018). Solid Digestion of Demineralized Bone as a Method To Access Potentially Insoluble Proteins and Post-Translational Modifications. *Journal of Proteome Research*, 17(1), 536–542.
- Gu, M., & Buckley, M. (2018). Semi-supervised machine learning for automated species identification by collagen peptide mass fingerprinting. *BMC Bioinformatics*, 19(1), 241.
- Ludwig, K. R., Schroll, M. M., & Hummon, A. B. (2018). Comparison of In-Solution, FASP, and S-Trap Based Digestion Methods for Bottom-Up Proteomic Studies. *Journal of Proteome Research*, 17(7), 2480–2490.
- Tsutaya, T., Mackie, M., Koenig, C., Sato, T., Weber, A. W., Kato, H., Olsen, J. V., & Cappellini, E. (2019). Palaeoproteomic identification of breast milk protein residues from the archaeological skeletal remains of a neonatal dog. *Scientific Reports*, 9(1), 12841.

Reviewers' Comments:

Reviewer #1:

Remarks to the Author:

I thank the authors for their thoughtful responses to my and the other reviewers' comments. I'm excited to see the adoption of optimized extraction and sample preparation in bone paleoproteomics especially with automation.

Something I failed to mention in the original review, I think the data analysis method will be beneficial in low-throughput species identification studies as well. Making the scripts available is very beneficial.

Line 95/Response to my review: While I agree that this approach can be used on a variety of high-resolution MS platforms that may exist in core facilities and reduces costs over traditional LC-MS/MS by time, I think the biggest limitation is the use of a very specialized LC that most facilities won't have. Can you clarify if the ultra-short gradients can be accomplished with other more commonly owned nanoLCs (nanoAcquity, Ultimate 3000, etc)? Is the same level of resolution possible on microflow instruments that are more typically used for high-throughput?

Data availability: Were the ZooMS spectra added to the ProteomeXchange dataset? If not please add them to a new dataset.

Reviewer #3:

Remarks to the Author:

The authors solved most concerns. The SPIN method is deserved to popularize. Some small questions as follows:

What kind of search software was used to identify peptides? Developed by the authors or just some commodity in line 107?

How long is the search time in species identification? A few minutes or tens of minutes?

Reviewer #4:

Remarks to the Author:

In this manuscript R  ther et al. present a new strategy, called SPIN, for species identification from archaeological material. They used for that state-of-the-art proteomics and innovative bioinformatical strategies to obtain a sensitive and specific method. They finally demonstrate that SPIN outperformed the standard method for this application, ZooMS, based on MALDI-TOF PMF. Although, the SPIN method uses expensive orbitrap instrumentation, the authors optimized the sample preparation for automation and used ultrafast LC gradients in order to reduce the cost of the analyses. Moreover, one can assume that Q-orbitrap instruments, already present in many MS core facilities around the world, will be more and more used for routine applications in the next years, therefore reducing their selling prices.

However, the authors claim the method could be applicable to any protein-containing material, which is overstated. In this study, an extensive curation of the public databases has been done regarding the 20 genes mostly found in bones, a matrix specific to these genes for 156 species has been created and specific peptides have been selected for fine grouping of the species. Although one can appreciate this impressive work, the translation of the method to another tissue or set of species would mean to provide this considerable effort again. For this reason, the sentence of line 78 should be modified and the title of the manuscript should mention that the method is specific to mammalian bones. It is fair to comment on a possible extension of the method to other tissue and species in the discussion on the condition of explaining that the sample preparation should be adapted and that a new gene selection

and alignment should be carried out.

According to figure 6, the method appears to be more specific and probably more sensitive than the PMF strategy. However, only a small subset of the 156 species in the model have been tested. While the study demonstrate that DirectDIA provide a lower amino acid coverage than DIA using spectral libraries, those libraries are available for only 13 species. Moreover, sequence information is incomplete for many species (as shown by figure S15) and the peptide selection for fine tuning has been done for 19 species only (PR201105 Manual SpeciesFineStructure Peptides.csv file). The study is anyway relevant, since the reference samples used in figures 3 and 6 are the most commonly found and studied. However, the authors should comment the points above in the discussion to keep the reader aware on the capabilities of the method for less frequent species.

The authors thoroughly described, in the supplementary information document, all the optimization steps including sample preparation and LC-MSMS analysis, which is helpful. However, only a small part of this extensive work is mentioned in the main text, some figures are even not cited. Few more sentences could be added to make to inform the reader on the information he could get in the supplementary information document.

In overall, the manuscript presents a considerable amount of work and a very innovative strategy which will be of high interest for the readers of Nature Communications. In addition to the points mentioned above, minor changes should be done to improve the clarity of the manuscript:

Main document

Line 182-186: Based on the legend of figure 2b, it seems that 177 species have been initially selected but only 156, having information on 14 genes or more, have been included in the final database. This information should appear in the text and not (or not only) in the figure legend. Can you also comment on the number of genes available for the species tested in this study? Is there information for all the 20 genes of those species? If yes, can we expect the SPIN method to be efficient on species with less gene information? And how the 14 genes threshold was chosen?

Fig 3d: It is not clear why the chimpanzee sample classified as *Pan paniscus* is displayed in blue on the graph which according to the legend correspond to correct species. However, as mentioned in the text this sample is assigned to the correct genus so it could appear in another color (not blue nor pink).

Line 247: 'we analyzed a set of 64 bones': there is 63 in the figure and in the legend.

Line 251: A table with the Salpetermosen samples identifiers and the corresponding morphological species assignment would be required. It could be a column in the Table S2.

Line 252: What is the rational on using high and low MS loading amounts? How much peptide material (μg) are in high and low load? Could it be possible that high load amount cannot be reached with very ancient bones?

Fig 4b is cited at the wrong place in the text (line 256)

Fig 4c contains many information and is not easy to understand. The colors scales for Identified sites and Relative Protease Intensity are contradictory since pink color represents high number of identified sites (expected for high intensity samples) but also represents High Relative Protease Intensity (expected for low intensity samples).

The authors should also comment on the 4 samples that are reported as 'signal too low' for the high

load but leads to a species assignment for the Low load.

Fig 4d is not commented at all in the manuscript. Moreover, it does not seem correlated to the information in Fig 4c. Please explain.

Fig 4e is cited as Fig 4d in the text.

Line 273: could you explain the meaning of "more degraded sample" and give a rough estimation of the degradation state of each sample in a table.

Fig5b: the layer 8 of Val boi is not mentioned in the text nor in the figure legend.

Line 306: According to figure 5b and after exclusion of 'below FDR' and 'weak signal' samples, it is not 20 out of 34 that could be assigned but 12 out of 34.

Line 391: the Salpetermosen samples are not cited in the sample description paragraph.

Supplementary information document

Please reorder the numeration of the supplementary tables (and if possible supplementary figures) as they appear in the main text.

Provide the tables in .xlsx format rather than .csv

In 'Reagent combinations for demineralization and extraction':

Supp Table 3:

The mention "successful LC-MS/MS" is imprecise. It should be removed from the legend of table S3. The LC-MS/MS successful should be removed from the table since the information on LC-MS/MS results is given by Fig S1.

Explain why the stage tips purified peptides have been measured at A280nm. Is it to inject the same peptide amount on the MS?

According to table S3, 3 protocols were successful: NP-40/EDTA/ACN; NP-40/HCl/ACN and NP-40/HCl/Heating but the last one does not appear on the fig S1. Please explain why you decided to not keep this one.

FigS3: replace axis name 'demineralization' by 'extraction time' and the replace the bar names '1hXTR and 2hXTR' by '1h and 2h'

Fig. S9: It would be helpful to provide a scheme representing the number and size of windows used in each method.

Information of the peak width for each LC method and cycle time of each DIA method would also be informative.

Fig S10: the y-axis is called 'absolute coverage' while the y-axis of the main figure 2d is called 'identified sites'. If both are representing the same measurement, the axis names should be harmonized (also true for Fig S12)

One reference is missing on page 32 in the deamidation section: "...protein preservation (?, ?)"

Fig S15: Replace 'both' in the legend by the corresponding databases (probably NCBI and Uniprot)

Fig S17: should be in the 'sample preparation optimization' section

Fig S1, S2, S3, S4, S5, S6, S9, S11, S12, S13 and S14 should be shortly mentioned in the main text.

The Methods section in the main manuscript does not contain all the details about the experiments described in the supplementary information document (Sample preparation protocols of the table S3 for instance or all the DIA parameters used for each method of the Fig S9). You could provide this information in a 'supplementary Methods' document.

Point-by-point rebuttal to REVIEWERS' COMMENTS

Manuscript: Nature Communications (NCOMMS-21-11457-T) by R  ther et al.

The reviewers' comments are provided in **black** font.

Our new responses to the reviewers' comments are provided below in **red** font.

REVIEWERS' COMMENTS

Reviewer #1 (Remarks to the Author):

I thank the authors for their thoughtful responses to my and the other reviewers' comments. I'm excited to see the adoption of optimized extraction and sample preparation in bone paleoproteomics especially with automation.

We are glad that our revised manuscript convinced the reviewer and we would like to acknowledge the useful feedback in the first revision that helped us get here.

Something I failed to mention in the original review, I think the data analysis method will be beneficial in low-throughput species identification studies as well. Making the scripts available is very beneficial.

Thank you for the positive feedback. The new (site-based) approach of palaeoproteomics data analysis has proven to be particularly useful for non-tryptic degraded peptides in our hands and we are looking forward to seeing it adapted by other groups.

Line 95/Response to my review: While I agree that this approach can be used on a variety of high-resolution MS platforms that may exist in core facilities and reduces costs over traditional LC-MS/MS by time, I think the biggest limitation is the use of a very specialized LC that most facilities won't have. Can you clarify if the ultra-short gradients can be accomplished with other more commonly owned nanoLCs (nanoAcquity, Ultimate 3000, etc)? Is the same level of resolution possible on microflow instruments that are more typically used for high-throughput?

We added the following sentence about alternative instrumentation: "As an alternative to using the latest LC-MS/MS equipment, the data acquisition methods required for SPIN should theoretically be transferable to previous generations of Orbitrap MS instruments^{29,30} and lower sensitivity microflow LC systems³¹ if compensated with higher peptide loads."

Data availability: Were the ZooMS spectra added to the ProteomeXchange dataset? If not please add them to a new dataset.

Yes, the ZooMS spectra are already part of the current ProteomeXchange dataset as peptide lists in ".txt" format. Having read between the lines, we absolutely agree with the reviewer that there is a lack of data sharing in the ZooMS community.

Reviewer #3 (Remarks to the Author):

The authors solved most concerns. The SPIN method is deserved to popularize.

We thank the reviewer for the positive feedback.

Some small questions as follows:

What kind of search software was used to identify peptides? Developed by the authors or just some commodity in line 107?

We added the two search engines used in this study as examples: ", such as Spectronaut²⁷ and Maxquant³⁴,"

How long is the search time in species identification? A few minutes or tens of minutes?

Since the search time is heavily dependent on the respective PC or server hardware, we unfortunately cannot give a reliable answer. Instead, we would like to point the reviewer to the two cited papers, which contain details about the search engines and computer hardware requirements.

Reviewer #4 (Remarks to the Author):

In this manuscript R ther et al. present a new strategy, called SPIN, for species identification from archaeological material. They used for that state-of-the-art proteomics and innovative bioinformatical strategies to obtain a sensitive and specific method. They finally demonstrate that SPIN outperformed the standard method for this application, ZooMS, based on MALDI-TOF PMF. Although, the SPIN method uses expensive orbitrap instrumentation, the authors optimized the sample preparation for automation and used ultrafast LC gradients in order to reduce the cost of the analyses. Moreover, one can assume that Q-orbitrap instruments, already present in many MS core facilities around the world, will be more and more used for routine applications in the next years, therefore reducing their selling prices.

We would like to thank the reviewer for this excellent summary of our key findings.

However, the authors claim the method could be applicable to any protein-containing material, which is overstated. In this study, an extensive curation of the public databases has been done regarding the 20 genes mostly found in bones, a matrix specific to these genes for 156 species has been created and specific peptides have been selected for fine grouping of the species. Although one can appreciate this impressive work, the translation of the method to another tissue or set of species would mean to provide this considerable effort again. For this reason, the sentence of line 78 should be modified and the title of the manuscript should mention that the method is specific to mammalian bones. It is fair to comment on a possible extension of the method to other tissue and species in the discussion on the condition of explaining that the sample preparation should be adapted and that a new gene selection and alignment should be carried out.

In the lines 77 – 79, we referred to the *concept* of the SPIN method, i.e. fast/automated sample preparation by PAC, rapid data acquisition, and site-based peptide data interpretation. We do not believe that our wording implies that the current SPIN implementation would be ready for other sample types without adjustments.

Since we cannot rule out the possibility of proteinaceous materials without species information, such as highly repetitive and conserved proteins, we changed the wording to "...applied to many protein-containing materials, ...".

We are very hesitant to changes of the manuscript title. Our motivation for a concise and broad title is reaching a wide audience with the main goal being the adaption of the SPIN workflow for species identification of non-mammalian species and other materials than bone. This was also our incentive for targeting a broad scientific journal.

To make the scope of our method clearer, we changed line 27 in the abstract: *"Here, we introduce "Species by Proteome INvestigation" (SPIN), a shotgun proteomics workflow for analyzing archaeological bone capable of querying over 150 mammalian species by liquid chromatography-tandem mass spectrometry (LC-MS/MS)."*

According to figure 6, the method appears to be more specific and probably more sensitive than the PMF strategy. However, only a small subset of the 156 species in the model have been tested.

We added the following sentence in the discussion to highlight this important point: *"It needs to be noted that the species selection in the test set was not aimed at broad coverage of mammalian taxa but rather at the more challenging task of discriminating a few common species from their close relatives."*

While the study demonstrate that DirectDIA provide a lower amino acid coverage than DIA using spectral libraries, those libraries are available for only 13 species. Moreover, sequence information is incomplete for many species (as shown by figure S15) and the peptide selection for fine tuning has been done for 19 species only (PR201105 Manual SpeciesFineStructure Peptides.csv file).

Thank you for catching this important point that we missed in the figure caption of S15. We made the following change: *"Taxa with less than 15 genes in the database were excluded in the species identification algorithm and are indicated by grey font."*

The study is anyway relevant, since the reference samples used in figures 3 and 6 are the most commonly found and studied. However, the authors should comment the points above in the discussion to keep the reader aware on the capabilities of the method for less frequent species.

We added the following text describing the current availability of reference data:

“Depending on the research question and archaeological context, studies may require an extension of the protein database and the set of spectral libraries. Although genomes are available for a plethora of extant mammals⁵⁰, the main bottleneck towards a more comprehensive protein database in our eyes is the genome annotation and sequence prediction for understudied species. With the appropriate sequence database and reference samples at hand, generating spectral libraries is a relatively straightforward task if required for a project. We envision that the numbers of available reference data will grow with more research groups sharing their protein sequence databases and spectral libraries, in the future.”

The authors thoroughly described, in the supplementary information document, all the optimization steps including sample preparation and LC-MSMS analysis, which is helpful. However, only a small part of this extensive work is mentioned in the main text, some figures are even not cited. Few more sentences could be added to make to inform the reader on the information he could get in the supplementary information document.

We are pleased to hear that the reviewer found the supplementary information useful. We added supplementary figure references as indicated, below.

In overall, the manuscript presents a considerable amount of work and a very innovative strategy which will be of high interest for the readers of Nature Communications.

In addition to the points mentioned above, minor changes should be done to improve the clarity of the manuscript:

Main document

Line 182-186: Based on the legend of figure 2b, it seems that 177 species have been initially selected but only 156, having information on 14 genes or more, have been included in the final database. This information should appear in the text and not (or not only) in the figure legend. Can you also comment on the number of genes available for the species tested in this study? Is there information for all the 20 genes of those species? If yes, can we expect the SPIN method to be efficient on species with less gene information? And how the 14 genes threshold was chosen?

To clarify the “15 gene cutoff”, we modified the following sentences: *“Further manual refinement was needed to remove faulty sequence inserts or obvious prediction errors like frameshifts and we removed 21 species that lacked more than 5 out of the 20 genes from the 177 available taxa, because we sporadically observed false identifications of species (data not shown). For all species identified in this study, sequences for all 20 genes were available for all taxa, except the white-tailed deer (19 genes), the European bison (15 genes), and the aurochs (15 genes).”*

Fig 3d: It is not clear why the chimpanzee sample classified as *Pan paniscus* is displayed in blue on the graph which according to the legend correspond to correct species. However, as mentioned in the text this sample is assigned to the correct genus so it could appear in another color (not blue nor pink).

We updated the figure legend of 3d: *“Correctly identified genus is highlighted in blue.”*

Line 247: ‘we analyzed a set of 64 bones’: there is 63 in the figure and in the legend.

Thank you for catching this. Fixed in the revised manuscript.

Line 251: A table with the Salpetermosen samples identifiers and the corresponding morphological species assignment would be required. It could be a column in the Table S2.

This was indeed a very relevant omission and we are happy that the reviewer caught this. We have added another column “MorphologyGroup (Species)” to table S2. The table description in the SI has been updated accordingly: *“For the bones from the Salpetermosen site in Denmark that were analyzed morphologically, the morphology-based species group and most likely species are listed in the “MorphologyGroup (Species)” column.”*

Line 252: What is the rationale on using high and low MS loading amounts? How much peptide material (μg) are in high and low load? Could it be possible that high load amount cannot be reached with very ancient bones?

We rephrased this section in our manuscript to make it clearer: *“Each specimen was morphologically analyzed by an experienced zooarchaeologist and the SPIN analysis was conducted in technical duplicates starting on bone powder level. Variations in the input amount, peptide recoveries, and LC-MS/MS performance resulted in one experiment with higher and one with lower average MS intensity.”*

Unfortunately, peptide quantification is rather inaccurate for ancient samples due to hydrolysis and degradation of aromatic amino acids, and we would prefer not to provide μg amounts. Similarly, peptide recovery from ancient bone is hardly predictable with reasonable precision. Spectroscopic pre-screening methods exist, but we have not applied them in this study.

Fig 4b is cited at the wrong place in the text (line 256)

The figure references have been fixed in the revised manuscript.

Fig 4c contains many information and is not easy to understand.

Although we are aware of the complexity of figure 4c, we prefer to display our species identification results as transparently as possible. In our eyes, reducing the results to “right” or “wrong” would take away the possibility of identifying multiple possible species with SPIN if distinguishing sites are missing. We hope that sorting the samples by species (morphology-based), which was done in the first revision, improved the readability.

The colors scales for Identified sites and Relative Protease Intensity are contradictory since pink color represents high number of identified sites (expected for high intensity samples) but also represents High Relative Protease Intensity (expected for low intensity samples).

We revised figure 4c by aligning the color scales. Besides being more aesthetical benefits, remaking the figure revealed a few critical mistakes that happened during the first revision and caused the inconsistencies between 4c and 4d. These are now fixed and figure 4c reflects exactly the contents of the result table S1.

The authors should also comment on the 4 samples that are reported as ‘signal too low’ for the high load but leads to a species assignment for the Low load.

We added the following sentence: *“We observed four cases of missing identifications in the high intensity replicate that were identified in the low intensity experiment, which we attributed to variability in the bone chips, input amounts, and peptide recovery.”*

Fig 4d is not commented at all in the manuscript. Moreover, it does not seem correlated to the information in Fig 4c. Please explain.

The main purpose of 4d was having a simple overview that distinguishes between cattle and broader bovine identifications. The following sentence was added to the manuscript: *“Nevertheless, we were interested in the performance of SPIN for distinguishing Bos and Bison in degraded material and therefore looked at the overall species distribution in the high intensity replica experiment (Fig. 4d). Discriminating between Bos and Bison was only possible for 5 out of 14 bovine bones and became significantly more challenging with lower sequence coverage (Table S1).”*

Fig 4e is cited as Fig 4d in the text.

Fixed.

Line 273: could you explain the meaning of “more degraded sample” and give a rough estimation of the degradation state of each sample in a table.

In this sentence, we were referring to the relative degradation state compared to the reference samples. To make this clearer, we added *“Compared to the well-preserved reference samples,...”*.

Determining the degradation state of individual samples was not subject of this study. General information can be found in the cited literature about the Salpetermosen site.

Fig5b: the layer 8 of Val boi is not mentioned in the text nor in the figure legend.

We modified the paragraph about deer identifications to include all 3 layers: *“The vast majority of the identified bones from layers 6 and 7 and all bones in layer 8 were classified as deer”*

Line 306: According to figure 5b and after exclusion of ‘below FDR’ and ‘weak signal’ samples, it is not 20 out of 34 that could be assigned but 12 out of 34.

Thank you for bringing this glitch to our attention. We fixed it in the revised manuscript: *“Finally, for Gruta da Companheira 12 out of 34 samples could be assigned a confident species identification.”*

Line 391: the Salpetermosen samples are not cited in the sample description paragraph.

The information has been added in the revised manuscript with the following sentence: *“Access to the collection of 63 bone fragments from the “The collection of 63 bone fragments dated to the Scandinavian Iron-Age was sourced from the “Salpetermosen Syd 10” (MNS50010) site in Denmark52.”*

Supplementary information document

Please reorder the numeration of the supplementary tables (and if possible supplementary figures) as they appear in the main text.

The order of SI tables has been fixed.

Provide the tables in .xlsx format rather than .csv

All tables are now provided in .xlsx format.

In ‘Reagent combinations for demineralization and extraction’:

Supp Table 3 (now S1):

The mention “successful LC-MS/MS” is imprecise. It should be removed from the legend of table S3. The LC-MS/MS successful should be removed from the table since the information on LC-MS/MS results is given by Fig S1.

We revised the legend of table S3 (now S1):

Table S3 | Combinations of demineralization and protein extraction agents. Bone powder was prepared from a Pleistocene mammoth bone fragment and 10 mg were used for testing each combination of demineralization and extraction agents. Reagent final concentrations given as percentage by volume or molarity. Stability of the combined demineralization solution and successful bead aggregation were assessed visually. Combinations without precipitation of reagents or insoluble bone minerals marked with a "+" in the "Stable Combination" column. Successful aggregation of paramagnetic beads during PAC marked with a "+" in the "Bead Aggregation" column, if the beads clumped together and the liquid phase did not separate into organic and aqueous phases. Successful protein capture as determined by peptide yield > 2 µg (absorption at A_{280 nm}, data not shown) and > 1000 peptide identifications by LC-MS/MS (Fig. S1) is marked with "+" in the "Protein Capture" column.

Explain why the stage tips purified peptides have been measured at A280nm. Is it to inject the same peptide amount on the MS?

We added the following sentence to the supplementary information: *“Spectrophotometrical peptide yields (A_{280 nm}) were used for rough estimation of peptide recoveries and adjusting the LC-MS/MS injection amounts. However, the results were very unreliable due to the low amounts of aromatic amino acids in bone proteins and are therefore not included, in the study.”*

According to table S3, 3 protocols were successful: NP-40/EDTA/ACN; NP-40/HCl/ACN and NP-40/HCl/Heating but the last one does not appear on the fig S1. Please explain why you decided to not keep this one.

We added the following sentence to the supplementary information: *“The combination of HCl demineralization with NP-40 extraction and aggregation by heat was excluded, because the spectrophotometrical analysis suggested that no peptides were present.”*

FigS3: replace axis name ‘demineralization’ by ‘extraction time’ and the replace the bar names ‘1hXTR and 2hXTR’ by ‘1h and 2h’

Fixed.

Fig. S9: It would be helpful to provide a scheme representing the number and size of windows used in each method. Information of the peak width for each LC method and cycle time of each DIA method would also be informative.

We added supplementary table S9, which contains these data. Furthermore, we added the following paragraph to the SI: *“To preserve high mass identifications without losing too much time, we added*

three significantly wider DIA isolation windows at the high m/z end of the overall MS2 range (770 - 970 m/z). Within the lower mass range, four different window counts were compared with the lowest window count only tested with the short and the highest count only tested with the long gradient. The resulting window widths can be found in table S9 and the complete MS methods are available in the uploaded raw data."

Fig S10: the y-axis is called 'absolute coverage' while the y-axis of the main figure 2d is called 'identified sites'. If both are representing the same measurement, the axis names should be harmonized (also true for Fig S12)

Changed all to "identified sites".

One reference is missing on page 32 in the deamidation section: "...protein preservation (?, ?)"

Fixed

Fig S15: Replace 'both' in the legend by the corresponding databases (probably NCBI and Uniprot)

Fixed

Fig S17: should be in the 'sample preparation optimization' section

Fair point. We delightfully updated the numbering of all supplementary figures in the SI, main text, and file names.

Fig S1, S2, S3, S4, S5, S6, S9, S11, S12, S13 and S14 should be shortly mentioned in the main text.

We added the requested supplementary figure references in revised the manuscript and reordered the figures as they appear in the main text.

The Methods section in the main manuscript does not contain all the details about the experiments described in the supplementary information document (Sample preparation protocols of the table S3 for instance or all the DIA parameters used for each method of the Fig S9). You could provide this information in a 'supplementary Methods' document.

As mentioned above, we added the DIA parameters in the added supplementary table S9.

Furthermore, the full methods are available in the raw files provided with this study. Besides this, we only described deviations from the main SPIN protocol in the supplementary information, because the methods would otherwise be completely redundant. All in all, we are convinced that the provided information is enough to fully reproduce the presented experiments.